# Nanostructured Surface Finishing and Coatings: Functional Properties and Applications

**DOI:** 10.3390/ma14112733

**Published:** 2021-05-22

**Authors:** Ileana Ielo, Fausta Giacobello, Silvia Sfameni, Giulia Rando, Maurilio Galletta, Valentina Trovato, Giuseppe Rosace, Maria Rosaria Plutino

**Affiliations:** 1Institute for the Study of Nanostructured Materials, ISMN–CNR, Palermo, c/o Department of ChiBioFarAm, University of Messina, Viale F. Stagno d’Alcontres 31, Vill. S. Agata, 98166 Messina, Italy; ileana.ielo@ismn.cnr.it (I.I.); fausta.giacobello@ismn.cnr.it (F.G.); ssfameni@unime.it (S.S.); 2Department of Engineering, University of Messina, Contrada di Dio, S. Agata, 98166 Messina, Italy; 3Department of Chemical, Biological, Pharmaceutical and Analytical Sciences (ChiBioFarAm), University of Messina, Viale F. Stagno d’Alcontres 31, Vill. S. Agata, 98166 Messina, Italy; girando@unime.it (G.R.); mgalletta@unime.it (M.G.); 4Department of Engineering and Applied Sciences, University of Bergamo, Viale Marconi 5, 24044 Dalmine (BG), Italy; valentina.trovato@unibg.it

**Keywords:** halochromic wearable sensors, drug delivery systems, flame retardant, industrial coatings, filtration membranes, antifouling coatings

## Abstract

This review presents current literature on different nanocomposite coatings and surface finishing for textiles, and in particular this study has focused on smart materials, drug-delivery systems, industrial, antifouling and nano/ultrafiltration membrane coatings. Each of these nanostructured coatings shows interesting properties for different fields of application. In this review, particular attention is paid to the synthesis and the consequent physico-chemical characteristics of each coating and, therefore, to the different parameters that influence the substrate deposition process. Several techniques used in the characterization of these surface finishing coatings were also described. In this review the sol–gel method for preparing stimuli-responsive coatings as smart sensor materials is described; polymers and nanoparticles sensitive to pH, temperature, phase, light and biomolecules are also treated; nanomaterials based on phosphorus, borates, hydroxy carbonates and silicones are used and described as flame-retardant coatings; organic/inorganic hybrid sol–gel coatings for industrial applications are illustrated; carbon nanotubes, metallic oxides and polymers are employed for nano/ultrafiltration membranes and antifouling coatings. Research institutes and industries have collaborated in the advancement of nanotechnology by optimizing conversion processes of conventional materials into coatings with new functionalities for intelligent applications.

## 1. Introduction

Different nanostructured coatings and surface finishing, characterized by length range between 1 and 100 nm, may be deposited on the external area of a matrix to implement or enhance the materials efficiency for several applications [1]. Actually, most of the atoms, by aggregating, may contribute to change the size of the grain boundaries thus improving the physical properties of nanostructured materials; this is in accordance with the Hall–Petch reinforcement method, by which the change in grain size can affect the number of dislocations accumulated at the grain boundary and the yield strength. [2]. Conventional materials have followed through the evolution of human civilization by defining the historical ages, as for stone, iron, bronze ages, up to the most recent cement, and silicon ages. Scientific and technological developments allowed the definition of a “new époque” in which materials acquired a more advanced meaning and specific functionalities. During the last century, the synergism between several scientific disciplines such as chemistry, physics, nanotechnologies, computer science, engineering, electronics, medicine led to innovative composite, nano, and smart materials that represent a step forward with respect to conventionally intended ones. This review introduces the study of nanostructured surface finishing and coatings, focusing in particular on smart materials, drug-delivery systems, flame-retardant, industrial, nano/ultrafiltration and antifouling coatings; all subject are described in depth through subsections reporting the synthesis, the processing and characterization of nanostructured surface finishing and coatings. The purpose of this review is to give a panoramic view of functional properties and the related applications of all kinds of nanostructured coating.

## 2. Advance in Halochromic Smart Textiles

### 2.1. Smart Materials: An Introduction

Smart materials are defined as materials able to respond to physical, chemical, or mechanical external stimuli by producing a quantifiable simple signal and combining the main functions of sensors, actuators, or transductors [3]. They were defined for the first time in 1989 in Japan, although the first examples coincided with the discovery of the shape memory materials and intelligent polymeric gels in the 1960s and 1970s, respectively. Besides the sensing properties, the stimuli responses could consist of actuations or corrective functions among which healing or self-mending are of great interest in biomedical applications. The most impressive nanotechnological development provides smart materials, including chromogenic, electrostrictive, magnetostrictive, piezo electric, thermoelectric materials and shape-memory alloys [4] (Figure 1).

Moreover, nanotechnological coating has fascinating and influential implications with protective actions in military and civilian fields as well as thermochromic, electrochromic and gasochromic smart materials, controlled drug release membranes, sensors for volatile organic compounds (VOC), contaminants, radiation, chemical warfare gases, land mines and smart coatings for adsorption of proteins [3]. The advancement in nanotechnology and the interdisciplinary activities between research institutes and industries has led to the optimization of increasingly technological processes for the conversion of conventional materials to design an ever-growing range of coating with novel functionalities for smart applications. Biosensing materials, pH-sensitive polymers, and thermo-responsive materials are examples of such intelligent materials, which find application in medicine, healthcare, fitness, military, professional workwear, and many other fields.

A large number of sensing molecules can be integrated with fabrics allowing the use of such smart fabrics in medicine, health care, fitness, wellness, diagnostic, and environmental fields [5,6]. Such integration can be possible through the development of smart (or active) coatings able to sense and react to specific changes in the environment thanks to reversible or irreversible variations in their physical or chemical properties [7]. Phase-change and memory materials, conductive, antimicrobial and chromic molecules, nanoparticles, are only a small section of the “active” molecules that can be integrated into coatings. Therefore, the designed smart coatings can react to external stimuli like humidity, heat, light, chemicals, pressure, and biomolecules thus providing large discontinuous changes in their physical properties [8]. The use of nanostructured coatings takes advantage of the precise control of their thickness at a nanometer scale. According to Bratek-Skicki [9], the rapidly-growing field of nanostructured stimuli-responsive coatings resulted in many advances in developing multi-functional systems. The coating responses can be color changes, energy release, shape adaption, drug release, changes in the dimensional structure of the textile fibers, and many others thus providing a wide panorama of wearable sensors as a consequence of their integration with textile structures. For example, through the use of deformation-sensitive fibers also in combination with conductive polymeric composite it is possible to obtain deformation sensors for respiratory rate monitoring, posture, or body movements [10]. Similarly, carbon nanotubes, conductive polymers or composite can be used to functionalize textiles to obtain moisture wearable sensors based on variation in the electrical resistance as a function of environmental humidity levels [11] or sensors for heart rate monitoring [12]. Advanced sensing nanomaterials for several applications such as solar energy harvesting, biomedicine, drug delivery, catalysis, spectroscopy can be engineered by using plasmonic nanoparticles. Among several examples, the combination of such plasmonic assemblies with stimuli-responsive molecules leads to reversibly reconfigurable plasmonic nanomaterials such as thermally responsive hydrogels, target-responsive biomolecules, and molecular photoswitches; in the presence of external stimuli, the plasmonic nano assembly provides changes in geometry or conformation [13]. Similarly, the conjugation of 2D materials and plasmonic nanoparticles leads to plasmonic composites showing non-linear optical properties whose spectral responses can be reconfigured in response to external stimuli and characterized by enhanced light harvesting with respect to the singular 2D counterpart thus making them interesting for sensing and spectroscopic applications [14]. The plasmonic assemblies represent reconfigurable platforms for flexible and stretchable electronics, where the 2D structure can be actuated to form a 3D structure by providing a versatile and dynamic platform with hybrid functionalities not shown by the 2D materials [14].

The change in phase of materials is a very attractive feature for developing innovative photonic device controlled by stimuli with tunable optical properties [15]. In this field, phase-change materials (PCMs), thanks to their reversible phase transition, data storage, high resistance and switching speed, are of great interest for developing metamaterials for optoelectronic devices, metasurfaces, rewritable optical data storage [15]. Among many non-volatile PCMs, chalcogenide materials are characterized by reliable and reproducible phase transitions, from amorphous to crystalline in response to stimuli which differ significantly in optical and electrical properties. One of the main representative chalcogenide PCMs is GeSbTe, which can reversibly switch between amorphous and crystalline phase in response to thermal excitation at high speed [16]. Commonly, the amorphous phase undergoes shifts to covalent bonding contrary to the crystalline phase that goes through resonant bonding [16]. Such bonding modes variations in PCMs provide a clear difference in optical properties: higher optical dielectric constants have been observed for crystalline PCMs than amorphous ones due to their resonant bonding [16]. Their reversible phase transition, non-volatile electrical/optical switchability, optical non-linearity, infrared transparency, photosensitivity make them interesting for developing reconfigurable metamaterial-based nanodevices with tunable optical properties with applications in solar cells, photonics, memories, and sensors [16].

### 2.2. Textile-Based Smart Materials

An important contribution to the advancement of smart materials was provided by an innovative way to consider the textile polymers besides their conventional use. During the last few decades, the fascinating and unique properties of textile materials such as flexibility, breathability, mechanical strength, simplicity of processing, biocompatibility (e.g., natural fibers) and washability make them interesting as substrates for the integration of smart functionalities, by paving the way for the research field dealing with the so-called smart textiles. Following the definition of smart materials, they are defined as textiles able to sense external stimuli, react, and adapt to them through specific functionality integrated with the fabric structure [17]. Flexibility, low weight and comfort are the main challenges in developing wearable sensors, which make them interesting and fascinating over the conventional bulky and rigid sensors. During the last decades of 1900, sensors and specific functionality started to be integrated with garments by obtaining prototypes of smart textiles for the long-term monitoring of physiological parameters in a non-invasive way and in real time. With this aim, new processes have been developed to provide strong synergies between textiles and electronics or between different scientific sectors by developing innovative hybrid technologies.

Knitted or woven fabrics with integrated functionalities for sensing, processing, and actuation (e.g., gestures, wearer’s postures, vitals tracking) were integrated with clothes. By referring to a generic process, the transfer of specific functionalities by textiles processes (e.g., weaving) or by integration with technological components (e.g., light-emitting diode (LED)) by preserving the main characteristics of textile fabrics is possible through “textilification” [18]. Similarly, a large number of sensors can be integrated with textile fibers, like ECG (electrocardiography), EMG (electromyography) and EEG (electroencephalography) sensors, thermocouples, luminescent elements, but also carbon electrodes, useful for detecting biomedical and environmental components and/or parameters (e.g., oxygen, salinity, moisture or contaminants) [19]. Therefore, iconic smart materials, such as electromyography pants, electroencephalogram caps, and electrocardiography T-shirts, have been developed for the detection of different health parameters [20]. An important issue to be addressed in the design of smart textiles is the compliance with the body deformations, of major importance for sensors for the monitoring of body temperature, fluids (sweat, tears or urine), volatile molecules (odors or breath composition) related to body fluids. Such an issue is mainly related to the integration of electronics with textiles which represent a limiting factor. Indeed, characteristics like flexibility (to accommodate body deformations), miniaturization, biocompatibility (to not affect daily activities) as well as low-consumption work capacity are the most desired in wearable smart sensing. Among many sensors, electronic elements, electroconductive fibers/yarns, miniaturized circuits conventionally integrated with textile fabrics through nanotechnological approaches, the manufacture of miniaturizing silicon sensors thanks to Si technology is of great interest in smart textiles for the detection of contaminants (bacteria, heavy metals, pesticides).

The integration of smart functionalities in textile fabrics can be achieved by nanotechnological approaches ensuring no interferences with textile engineering processes and the maintenance of the intrinsic textile properties. Such nanotechnological routes for surface modification of fabrics to obtain smart textiles could be represented, among several examples, by electrospinning, grafting polymerization, and sol–gel approaches.

### 2.3. Textile-Based Optical pH Sensors

An interesting and innovative field in which the so-mentioned nanotechnologies were fully investigated is represented by wearable optical sensors, whose chromic and halochromic smart textiles are the main representatives. In the smart textile field, the optical, easily detectable and non-destructive signal provided by chromic materials is strictly related to the nature of the external stimuli and classified as photochromism (color change induced by light), thermochromism (color change induced by heat), ionochromism (color change induced by ions) [21] thus providing phosphorescence, fluorescence, or visible stimuli-dependent color changes. Stimuli like sorption of chemicals, radiations, temperature changes, presence of heavy metals, redox reactions, as well as pH changes could be easily detectable and monitored by smart materials thanks to the integration of functional molecules such as indicator dyes, pigments or their advanced hybrid systems.

Smart textiles based on optoelectronic systems for the detection of pH variation are fascinating and innovative over the conventional electrochemical pH electrodes. With respect to the latter, optical pH sensors ensure a longer lifetime, low cost, safety, mechanical robustness, fast and reversible response, as well as the possibility of miniaturization and high signal to noise ratio (SNR) [22]. On the other hand, ambient light and some sweat components could affect the sensor selectivity, and the long-term stability of the sensor can be influenced by the leaching or photobleaching of the sensing halochromic molecule.

The monitoring of the sweat pH is of great interest since the composition of such biological transparent fluid gives useful information concerning human health. Due to the main role of the sweat pH in body hydration and mineralization levels, in skin pathologies, the real-time monitoring of sweat pH provides information dealing with corrective approaches for re-hydration or skin disease with applications in fitness, healthcare, medical diagnostic fields. Moreover, an increase in the sweat pH values is correlated with an increase in the sweat rate, as well as the concentration of some electrolytes, such as Na^+^: the higher the Na^+^ concentration in the sweat, the higher the sweat pH [23,24].

#### 2.3.1. Development Methods of Halochromic Coatings for Optical Textile-Based pH Sensors

To develop a reversible, robust and optical chemical sensor, the stable immobilization of an indicator dyestuff onto textile surfaces should be ensured in order to avoid dye leaching. According to the nature of both textile surfaces and chromic smart materials, intermolecular forces, ionic or covalent bonds or weak interactions (e.g., hydrogen bonds, dipole-dipole, or induced dipole–dipole interactions) between textile fibers and smart coatings can be established to ensure the adhesion of the functional materials [25].

Conventional dyeing methods, including exhaustion and impregnation processes, are low-cost and facile approaches for the introduction of chromic or halochromic molecules in textile polymers and based on the diffusion of dye molecules into fibers or on interaction between the dye and the fiber, respectively. Both of them are characterized by some disadvantages consisting of elevated temperature, low exhaustion, and fixation levels, insufficient affinity between dye and fibers, long dyeing time, high dye leaching, and slow response rate. As previously mentioned, several synthetic strategies can be carried out for the stable immobilization of sensing molecules in textile substrates, among which electrospinning, grafting polymerization and sol-gel technique.

##### Electrospinning

Electrospinning approach represents an efficient alternative to conventional dyeing technique for the incorporation of dyestuff into the textile structure thus ensuring a low dye leaching as a consequence of covalent interactions between dye and fibers [26]. Moreover, with respect to conventional dyeing methods, electrospinning provides several advantages such as low-process time and energy [26]. The halochromic functionalities can be introduced during the electrospinning of nanofiber, thus obtaining stable chromic and halochromic functionalities in textiles and combining the intrinsic properties of nanofibers and pH-sensing functionalities [26]. By starting from different polymer solutions or melted forms of polyurethane, nylon, polyacrylonitrile polyvinyl alcohol, or polylactic acid, and applying an electric field it is possible to obtain nanofibers with diameters between 10 and 500 nm [27]. By tailoring ambient and solution parameters an interconnected membrane-like network of mat fibers can be achieved. Therefore, via electrospinning of monofilaments, uniform and ultra-thin coating of nanofibers with different properties can be deposited onto textile fibers [27]. In particular, controlling the time processes makes it possible to tune the coating thickness to ensure the highest sensor sensitivity [28]. Van del Schueren et al. [29] provide an interesting example of nitrazine yellow (NY) pH-sensitive polyamide (PA) (6 and 6.6) nanofibrous non-wovens synthesized by electrospinning. In particular, the dye was added directly before the nanofiber formation thus obtaining a highly effective halochromic system with a fast response. Indeed, the performances of the halochromic sensor take advantage of the use of nanofibers since their high porosity and specific surface area enhance the response time and the sensitivity of the halochromic sensor itself. Experimental results demonstrated that the introduction of the NY dyestuff during the electrospinning of PA did not affect: (i) the process; (ii) the morphology and average diameter of the obtained fibers; (iii) the NY halochromism of electrospun PA. Moreover, the typical NY color transition from yellow to blue for increasing pH, as well as the fast response (5 min vs 20 min of conventional dyed PA) were provided by the nanofibrous non-wovens. It was also demonstrated that the nature and the structure of polyamide fibers influence the halochromic response thus observing less bluish color for PA 6.6 compared to PA 6 due to different interactions between dye-fibers and fiber accessibility [30]. However, all textile samples revealed reliable color changes as a function of pH variation thus highlighting the potentiality of the electrospinning process and of polyamide as potential parameters for colorimetric textile sensors.

##### Grafting Polymerization

Another interesting chemical approach for the stable immobilization of halochromic molecules in textile structures is represented by the grafting polymerization, which represents a widely investigated route for the surface modification of materials through the introduction of coating with high durability. Such a process forecasts the use of reactive molecules, such as vinyl or acrylate monomers, able to provide, in proper conditions, free radical species for the generation of “active sites” onto textile surfaces according to initiation and propagation reactions. Such radical species can be activated by several processes like thermal grafting (by radical initiators such as persulfate ions) or high energy radiation (e.g., ultraviolet (UV) light) or by a high-energy system of ions, radicals and metastable molecules such as in plasma-induced grafting. According to the initiation process, different radical initiators and sources of energy must be properly chosen. Photo-induced grafting represents a valid technique to modify cellulose-based textile polymers [31,32] by grafting functional moieties on the outermost surface of the substrate without affecting the bulky structure due to the depth functionalization of about 200 micron in depth. As an example, according to photo-grafting process, the nitrazine yellow (NY) dyestuff was stably immobilized by means of glycidyl methacrylate (GMA) monomers onto cellulose fabrics with the aim to realize a robust and reliable wearable pH-sensor [33]. In a common mechanism, the triplet state species of the photoinitiator (benzophenone) generated by UV light, are able to abstract hydrogen atoms by the OH groups of the cellulose chain by forming cellulose radicals [34]. The latter can attack the double bond of the bifunctional GMA monomer whose epoxy ring group was previously catalytically opened for the covalent functionalization of the NY. A similar approach was performed by Trovato et al. [35] with a radical procedure designed to promote the *in situ* thermal-induced grafting of the GMA-functionalized NY. In this research, potassium persulfate (KPS) was used as a free-radical initiator able to dissociate giving potassium and persulfate ions under thermal heating [36]. Such a homolytic cleavage produces sulfate radical anions that generate further active sites on the cellulose surface by the abstraction of hydrogen atoms. Similar to photografting, the dyestuff-acrylate monomers can attack these active sites onto the cotton surface by promoting new covalent bonds between monomers and the textile surface thus resulting in a robust and reliable halochromic wearable sensor.

##### Sol–Gel Technique

In the design of stimuli-responsive coatings, the sol–gel technique was widely investigated thanks to the fascinating properties of the derived thin films, among which high chemical stability, optical transparency, flexibility, particular physical properties, high durability, mechanical and chemical protection towards fibers [37]. Not surprisingly, some applications deal with UV-protection, halochromic, antimicrobial, hydrophilic, self-cleaning, and flame-retardant coatings for textiles [38,39,40,41,42,43,44,45,46,47]. According to subsequent hydrolysis and condensation reactions of metal alkoxides (e.g., Ti, Si, Al, Sn, V or Zr salts), or organometallic substrates in which the metal is bonded to hydrolysable organic functionalities, hybrid organic–inorganic materials with three-dimensional porous structures and very pure at the atomic scale can be obtained by tailoring the synthetic parameters. Functional organic molecules (e.g., active principles, halochromic molecules, carbon nanotubes, drug molecules) can be stably immobilized in the hybrid network according to weak or covalent interactions by leading to smart coatings with appealing properties for sensing applications. The nature of the aforementioned interactions affects the sensing properties of the resulting smart materials: for instance, non-covalently linked molecules are advantageous for the design of controlled release coatings since the retention of such molecules is stimuli-dependent [48]; conversely, covalently bonded molecules provide robust smart materials for advanced sensing applications (e.g., pH-sensitive coatings). Among several sol–gel precursors, Si-based have aroused much interest in the design of smart coatings thanks to their lack of cytotoxic effects on human skin cells [49] although their low reactivity with respect to other metal alkoxides. Interestingly, several examples are reported in the literature concerning the use of (3-glycidoxypropyl) trimethoxysilane (GPTMS) due to its bifunctionality: by epoxy ring-opening present in its structure, organic molecules can be covalently linked and, simultaneously, thanks to the hydrolyzed methoxy silane functionalities in the 3D network, the crosslinking toward surface materials can be achieved. The latter can be advantageous in the design of halochromic wearable sensors, thus resulting in the lowest dye leaching from textiles with respect to conventional dyeing procedures.

Several examples are available in the literature dealing with the covalent immobilization of a halochromic molecule by epoxy ring-opening of the GPTMS that can be performed by acid or base catalyst according to the anion, cation, or covalent nucleophilic mechanism. In this regard, a detailed study was carried out by Guido et al. concerning the use of the Lewis acid BF_3_OEt_2_ as the catalyst in the covalent immobilization of NY by the epoxy ring-opening of GPTMS [50]. The sol–gel reactions were performed using three different concentrations of the Lewis acid (1%, 5%, and 10% *w*/*w* GPTMS) to investigate the effect of the acid concentration on both the ring-opening and the formation of the silanol network. The solution of NY-doped silica matrix was applied onto cotton fabrics and compared to textiles treated with reference GPTMS solutions. It was demonstrated that the concentration of Lewis acid is of great importance to tune both the catalyst and the cross-linker effect; the higher the amount of acid the higher the formation of a more crosslinked inorganic network useful for an efficient hybrid halochromic matrix fixation onto textiles structure. Such experimental findings confirm the increased washing fastness of textile treated with silica matrix obtained by the high concentration of catalyst and the typical pH-response of NY although its covalent immobilization with GPTMS by sulfonic groups and epoxy ring-opening thus demonstrating the effectiveness of the sol–gel techniques in the design of a robust and reliable wearable pH sensor [50].

The versatility of nanotechnological processes in the design of halochromic sensors is of great importance to ensure the deposition and the effectiveness of coatings onto textiles with different natures. In this regard, Van der Schueren et al. [26] described a sol–gel procedure for methyl red (MR) dyestuff stable immobilization by HCl-catalyzed epoxy ring-opening of the GPTMS for polyamide textiles (PA) by demonstrating the feasibility and efficacy of the halochromic hybrid composite respect to conventional dyeing techniques. The efficiency of the polyamide fabrics treatment was demonstrated in terms of washing fastness of the sol–gel treated PA samples that showed increased durability to launder cycles (reducing the dye leaching) and higher halochromic response compared to PA conventionally dyed with MR. Contrary to potential limiting factors of the sol–gel approach, such as the longer response time due to the diffusion of the molecules into the dried silica network, shorter and easier responses to pH changes were measured for sol–gel treated PA samples than those conventionally dyed (4 min vs. 20 min, respectively). Such a finding was proved by morphological images of the sol–gel PA treated cross-sections that reveal the presence of the halochromic molecules only on the surface of the fibers in opposition to the conventional dyed PA textiles in which the MR molecules penetrated inside the singular fibers resulting in longer responses.

As reported in the literature, the epoxy ring-opening of the GPTMS can also undergo alcoholysis, hydrolysis, or polyaddition reactions to provide diols and then β-hydroxy ethers or polyethylene oxide (PEO) 3D network. Even if the covalent linkages between a halochromic molecule and the silica matrix ensure stability in terms of low leaching of the dye molecules, it was demonstrated by Rosace et al. [51] that also the encapsulation of resorufin as indicator dyes in the GPTMS-PEO matrix according to weak interactions can provide a reliable wearable pH sensor. Notwithstanding the slight influence on the coating durability due to the weak interaction, low leaching of the resorufin from textiles was observed. Moreover, the efficacy of the non-covalent resorufin-PEO immobilization onto cotton fabrics and the maintenance of the typical color variation of the pure dyestuff in the pH range between 2 and 8 were demonstrated by ultraviolet–visible (UV-Vis) spectroscopic measurements.

To provide a reliable wearable pH sensor, the hybrid polymeric network should also ensure the maintenance of the dyestuff photostability in the visible light. Plutino et al. [52] provided a detailed study concerning the photostability of the methyl red dye covalently functionalized with GPTMS under UV-Vis radiations at pH 2 and 8. Experimental results demonstrated that, compared to the unfunctionalized dye, the silanized molecule shows an increased photostability due to the missed influence of the hydrogen bond on the electron density of the azo bond. Moreover, a substitution in the *ortho* position of the phenyl group relative to the azo bond by GPTMS enhanced the photostability of the functionalized MR dyestuff. As a consequence, the photodecomposition of the hybrid molecules by UV light was prevented since the reduced influence of the so-mentioned hydroxyl following the GPTMS functionalization that acts as a shield towards the hybrid halochromic molecule towards photodecomposition. Experimental results demonstrated both the potentiality of azo dyes and silica hybrid matrix in the development of halochromic materials ensuring the efficiency of the designed systems and the lightfastness [52].

Notwithstanding this, the deprotonation–protonation process of the immobilized dyestuff should be ensured by the silica matrix involved in the dyestuff functionalization but even though by the electronics integrated with for the real-time monitoring of the sweat pH.

In this perspective, several studies have been provided by Caldara et al. concerning the development of optoelectronic colorimetric pH sensors using methyl red or litmus as dyestuffs covalently linked in GPTMS matrices [53,54,55]. The optoelectronic device for the color sensing variations was realized with a LED and a photodiode to implement the light-to-frequency conversion as the spectral sensitivity curve of the photodiode is monotonic in the smart fabric color variation range. The as-designed optoelectronic system, besides wearability, can take halochromic reflectance intensity variations and is characterized by low power consumption and easy connection towards a microcontroller for the processing of data. The halochromic system realized through the GPTMS immobilization of Methyl Red coated onto cotton fabrics integrated with electronics (Figure 2) provides, in the pH range 4.0–6.0, a resolution greater than 0.05 pH units with a color reproducibility within 2%, a monotonic relation between the frequency output of the color sensor and the pH, and a settling time on H^+^ variations of few seconds and few minutes in wet and dried conditions, respectively [53].

In another study, the designed sol–gel smart composite was employed in the real-time monitoring of both sweat pH and skin temperature [54]. In this regard, the wearable sensor platform was developed by using litmus, a non-toxic dyestuff, in a GPTMS functionalization of textile, and a silicon-based thermometer. Besides the latter, the electronic device includes an RGB color sensor, a low-power microcontroller, a nonvolatile memory, a power supply and a Bluetooth miniaturized module. The smart textile was designed with the aim of minimizing the power consumption and performing high resolutions and accuracies thus measuring greater than 0.01 °C and ± 0.4 pH temperature resolution and accuracy, respectively. To evaluate the reliability of the designed smart platforms, the described system was tested five times on-body during a bike exercise by measuring the sweat simultaneously through the textile based wearable device and an electrochemical pH meter, used as reference (Figure 3). The high sensitivity and the embedded processing allow a resolution of 0.2 pH units and an accuracy equal to 0.5 pH unit to be obtained with respect to the reference pH-meter [55]. Moreover, the wearable platform reveals a settling time of 8 min due to the poor sweat rate at the beginning of the perspiration [55].

The as-obtained non-invasive wearable pH sensor, thanks to the on-body experiments performed [55], reveals fascinating potentiality in the monitoring of athletes’ parameters for the evaluation of body hydration level or other health parameters [54] with applications in sport, fitness and medicine fields.

In this panorama, “Health Belt” (Figure 4) is proposed as a prototype of halochromic smart textiles for sweat pH real-time monitoring. It consists of an elastic strip where a cotton fabric coated by a silica-based halochromic film was embedded as the sweat pH-sensing element. The colour-changing is collected by a miniaturized electronic device, used as a read-out system, integrated with the treated cotton sample.

As already stated, the involvement of the sol–gel technology ensures the adhesion of the halochromic film onto textiles thus providing an environmentally friendly and healthy, safe human application. The miniaturized electronic device through the LED/photodiode system is able to detect the pH-dependent colorimetric variation of the textile, transmitting the data remotely by a Bluetooth module.

The described nanotechnological processes highlighted the potentiality of smart materials in the development of advanced, reliable and robust wearable pH sensors. Even more innovative applications would be provided as results of the combination of smart coatings, textile fabrics and microelectronics.

## 3. Drug-Delivery/Release Nanosystems

### 3.1. Functional Coating for Drug-Delivery Nanosystems: An Introduction

Active pharmaceuticals are allowed, under specific conditions, to reach the target site and improve its effectiveness, through the use of controlled drug-release nanosystems [57]. Drug-delivery systems contribute to implementing specific properties of ‘free’ drugs by improving biodistribution, solubility in biological media and their in vivo permanence. By employing a nanosystem that incorporates or encapsulates drugs and releases them, under external stimulation such as pH or temperature, controlled release can be achieved [58]. Controlled-release or drug delivery systems are employed to obtain:constant release into the blood of quantities of therapeutic compounds, avoiding drug waste;repeatable and scheduled long-term release rates;reduction of side effects;personalized therapy;drug stabilization [59]

A proper excipient helps keep drugs intact until delivery and facilitates their release at desired sites with maximum efficiency, while an appropriate processing method avoids unwanted degradation and waste of nanosystems [60]. These systems also change disadvantageous pharmacokinetics of some ‘free’ drugs. In addition, extensive loading of pharmaceuticals’ administration on drug-delivery systems can distribute ‘drug reservoirs’, for controlled and continuous release, to provide the drug level within the therapeutic window. Currently, many delivery nanocarriers based on nanometric size compounds such as micelles, dendrimers, nanotube and metallic nanoparticles have been designed. A lot of research has been focused on these delivery systems to support and provide a promising alternative to chemotherapy. Pharmaceutical research is attracted to planning advanced antineoplastic drugs with specific selectivity on cancer cells. In recent years, chemotherapy has principally focused on destroying all rapidly proliferating cells. The disadvantage of this therapy is that the body’s other rapidly dividing cells, such as in the hair follicles and intestinal epithelium are also killed off, leaving the patient to cope with harmful side effects. These new drugs should be able to exceed any resistance of the tumoral cells and they should provoke bland side effects.

#### 3.1.1. Self-Assembled Polymers for Nanocarriers

Chemotherapy drugs, used in current therapies, are often charged into lipophilic self-assembled molecules so, micelles are also excellent carrier systems to make insoluble drugs soluble due to their hydrophobic core and hydrophilic shell [61]. The further functionalization of the micelle’s surface with PEG increases the ability of the nanocarriers to pass through tumoral tissue as a result of passive transport, therefore resulting in higher drug concentration in cancer cells. Numerous polymeric micelles, incorporating anticancer drugs, are under clinical trial [62] and others are already approved for breast cancer patients. Dendrimers are repetitively branched macromolecules with lots of functional groups available for the connection of drugs, which are used as targeting and imaging agents and their absorption, distribution, metabolism and elimination profile is correlated to diversified structural characteristics. Nanoparticle therapeutics based on dendrimers can improve the therapeutic index of cytotoxic drugs by using biocompatible segments, such as PEG, acetyl, glicosane and various amino acids, linked on the surface area [63].

#### 3.1.2. Functional or Coated Nanofillers

There are several other models of functional nanofiller which show promising results in cancer treatment, a system used today contemplates functionalized carbon nanotubes. This carbon nanosystem is an allotropic form of carbon with a cylindrical structure extending on a number of sheets in concentric cylinders (single-walled carbon nanotubes and multiwalled carbon nanotubes) [64]. Water-insoluble drugs can easily be loaded on the hydrophobic hollow interior of carbon nanotubes. The large surface area consents to a specific outer surface functionalization for defined cancer receptors as well as contrast agents [65]. A spherical molecule such as fullerene (C_60_) and its derivatives are evaluated for the treatment of cancer [66] thanks to its ability to enhance the cytotoxicity of chemotherapeutic agents [67]. A study conducted using doxorubicin uploaded on the complex of fullerene C_60_ demonstrated that tumor volumes of treated rats were 1.4 times lower compared to the untreated rats [68]. These results are probably due to the direct action of the C_60_ + doxorubicin complex on tumor cells as well as immunomodulating effect. The development of nanoparticles has contributed to a new route for chemotherapy. With the design of smart nanoparticles, targeted drug delivery at the tumor site or a specific group of cells widely prevent the toxic and unwanted effects on other healthy tissues and organs [69]. Gold nanoparticles take advantage of their unique chemical and physical properties to carry and release drugs, and they are able to deliver the different size of drug molecules from little one to large biomolecules such as peptides, proteins or nucleic acids like DNA or RNA. This nanosystems recognize the surface of anionic protein as a result of interdependent electrostatic interaction and inhibit its activity [70]. Gold nanoparticles could be functionalized with lots of molecules with appropriate functional groups, in the monolayer [70]. Carbazoles, for example, were widely investigated for all their properties, which can be improved by changing functionalized groups or by introducing suitable substituents on carbazole core, with the intent to gain new and unique properties as antioxidant or antimicrobial. These compounds promote antiproliferative activity and considerable apoptotic response approaching cancer cells selectively. An approach to release pharmaceuticals, as carbazoles, involves the use of gold nanoparticles. These functional coated nanocarriers are suitable to deliver different payloads into target cells. In addition to the surface chemistry of gold nanoparticles, a promising perspective, in cancer therapy, is the photothermal damage [71] to cancer cells, which is an additional technique for enhancing the selective damage of unhealthy cells, based on the irradiation of nanoparticles, with 20 ns laser pulses (λ = 532 nm), to produce local heating. Circumstances for effective drug-release therapy can be improved thanks to external stimuli such as light or from the inside, through variations in the pH levels [72]. Tunable size and functionality make them a suitable scaffold for the efficient delivery of biomolecules. It has also been demonstrated that functionalized gold nanoparticles can act as carriers of insulin [73] and chitosan, a green biopolymer stabilized them. Gold chitosan-coated nanoparticles strongly adsorb insulin on their surface and are efficient for transmucosal delivery of insulin. For *in vivo* applications, the target of nanocarriers is the diseased tissue after the release into the circulatory system. There are two methods to deliver nanocarriers, ‘passive’ targeting and ‘active’ targeting [74]. The first one depends on vectors’ release in unhealthy cell tissues due to extravasation through a cracked blood vessel. Thanks to the nanometric diameter, the nanocarrier systems take advantage of the enhanced permeation and retention (EPR) effect [75]. On the other hand, ‘active’ targeting holds ligands on the carrier surface, for distinct recognition by cell surface receptors. A combination of both types of targeting will distribute an ideal carrier for in vivo delivery. Nanocarriers experience a non-specific uptake and possible degradation in macrophages. Therefore, targeting is essential for maximizing drug efficiency as well as minimizing side effects. Different physicochemical properties, such as size, PEGylation, or the ligand choice, coordinate non-specific versus target-specific uptake [76]. Gold nanoparticles with or without PEGylation of varying sizes (50, 80, 100, or 150 nm) are used for active targeting of cancer cells. Generally, PEGylation increases blood circulation lifetime and a specific ligand should facilitate filtration of nanocarriers into target cells. Two targeting molecules, folic acid (FA) and methotrexate (MTX) are specifically recognized by folate receptors that are overexpressed on the surfaces of many tumor cells [77]. Gold nanoparticles are not only convenient for cell-specific targeting, but also for localization into desired organelles. Recent research [78] proved that PEGylated gold colloids, functionalized with adsorbed protein, better detect a nuclear localization signal, for delivering drugs into nucleus’ cells. Nanoparticles have been developed as a promising scaffold for implementing drug delivery release compared to traditional delivery vehicles. They combine low toxicity, high surface area and tunable stability provides them with unique properties such as bioavailability and nonimmunogenicity.

### 3.2. Smart Polymers for Drug Nanocarriers

Drug delivery nanosystems are obtained by employing functional biocompatible materials or smart polymers as capping matrices, sensitive to particular external physiochemical stimuli, able to release an active biomolecule in the target site and at an adequate rate in response to specific functions [79]. New smart polymers for the controlled delivery of therapeutic drugs have been developed in the field of polymer engineering. Their physical, chemical, and biological signals can be supported by external sources, employed as triggering stimuli, and they can be promoted by internal environment conditions [80]. Smart polymeric systems may be dissolved in aqueous solutions or may be chemically grafted onto aqueous-solid interfaces, chemically bonded through hydrogen-bond systems, or hydrogel formulations [81]. The desired pharmacological action is not achieved with the rapid release dosage forms and it is, therefore, very important to control the timing of drug release. The release times are established based on the nature of the drug: water-soluble drugs require a slower release and a longer duration of action; those lipophilic require an increase in solubility to have an adequate therapeutic level, those with a short half-life require repeated administration and, finally, those with indefinite action require delivery to the target sites. To receive the drug level required for treatment, the drug-delivery system should deliver exact amounts of a particular drug at a planned rate. Several factors must be considered to design and implement this system, such as the physico-chemical properties of the drug, its route of administration and its pharmacological and biological effects [82]. Some advantages of controlled release systems are (i) the maintenance of drug levels within the desired range, (ii) the reduction of side effects, such as toxicity and frequency of administration, (iii) the improved efficacy [83]. The problems encountered in the use of these systems are due to the possible toxicity of the materials used, the need to operate through surgical procedures to insert or remove the system, possible poor availability of the system and high production costs [82]. New polymeric materials are currently being formulated, which respond to specific environmental changes in biological systems [84]. *Stimulus* reactive polymers mimic the behavior of biological molecules, thanks to external stimuli or changes in the local environment that can trigger a change in properties, such as solubility, shape, conformation, charge and size. Drug release could be regulated in spatial conformation by targeting and in time as a function of external stimuli [85]. The smart polymers, currently under study, are classified into the following categories: temperature-sensitive, dual stimulus reactivity, phase, light and biomolecule sensitive.

#### 3.2.1. Temperature-Sensitive Smart Polymers

Temperature is an easily controllable condition, so temperature-sensitive smart polymers have practical advantages in vitro and in vivo [86]. Polymers with a low to medium critical solution temperature (T = 30/40 °C) commonly dissolve in aqueous solvents, while polymers with a higher critical solution temperature (T > 50 °C) better dissolve in organic solvents [87]. The drug may be released as a result of changes in body temperature following a fever or local infections. The sensitivity to temperature, in polymers, depends on the equilibrium between hydrophilic and hydrophobic portions, and these intermolecular and intramolecular interactions lead to aggregation of the polymer chain [80]. Poly (*N*-isopropyl acrylamide) (PNIPAAm), is the most studied intelligent temperature-sensitive polymeric system. The solubility in water decreases drastically when heated to approx. 32 °C in pure water and a few less in physiological saline solution [81]. Pluronic block copolymer is a thermosensitive polymer capable of releasing biomolecules (proteins or lipoproteins) as a function of temperature variations. Pluronic diblock copolymer microspheres have a porous and hydrophilic structure. The release rate of the biomolecules is modulated according to the temperature variations and the surface functionalization of the Pluronic copolymer [88]. Gelatin and carrageenan are natural polymers that show a sol–gel transition and adopt an amorphous spatial conformation in solution at high temperatures, while during cooling, a continuous and homogeneous network is formed. The solubility of a polymer in water depends on factors such as temperature, molecular weight or the addition of an additive or co-solvent. The variation of the balance between the intermolecular forces causes the disassembly of the micelles, thus freeing the encapsulated host. This response can be induced from within, by exploiting the tendency of tumor tissues to have a slightly higher temperature, or by applying heat externally. Poly(*N*-isopropylacrylamide) (PNIPAAM) undergoes an abrupt phase transition at 32 °C and separates from the aqueous phase. PNIPAAM is non-toxic, and can be chemically adapted by changing the alkyl part or by copolymerizing it with other more hydrophilic monomers but it is not biodegradable. There are natural temperature-sensitive systems and they are suitable carriers. For example, collagen with glycineproline- (hydroxyl) proline (Gly-Pro-Pro (Hyp)) forms a triple helix. The thermo-sensitive polymers can be classified according to the mechanism and the chemistry of the inner groups. These polymers solubilize many hydrophobic drugs such as paclitaxel and have an excellent formulation for poorly water-soluble drugs [80]. Micelles are formed when the concentration of the polymer is increased above the critical micellar concentration (CMC), then below the low critical solution temperature (LCST) the heat-sensitive polymer is hydrated and hydrophilic (Figure 5).

#### 3.2.2. Phase-Sensitive Smart Polymers

Release systems for biomolecules should preserve their biological activity and conformational stability. Intelligent polymer systems were tested that form in situ, injectable, phase-sensitive gels, potentially usable for drug delivery control. These injectable formulations allow single injections of high doses with small volumes and small needles and improve the stability of biomolecules. An instant gel forms in situ after these formulations are injected. The hydrophilic solvent causes the formation of a shell on the outside, while the hydrophobic solvent slows the penetration of water in order to decrease the hydrolysis of the polymer and potentially increases the stability of the biomolecules [89]. In these systems, drug release rates can be controlled by optimizing factors such as drug loading and solvent composition [90]. In drug-delivery systems, the response to light can be introduced through a linker that can be cleaved by irradiation with electromagnetic radiation having a certain wavelength. Encapsulated hosts are activated or released after being irradiated with a radiant source from outside the body, resulting in a spatial and temporal release [91]. These polymers are designed for use in microsystems, medical imaging and tissue engineering. Photo-refractive polymers with near-infrared (NIR) sensitivity have also been reported, such as a matrix of poly (*N*-vinyl carbazole), *N*-ethylcarbazole as plasticizer and 2,4,7-trinitro-9-fluorenone (TNF) as a sensitizer. These NIR-sensitive materials could be used to visualize tumors thanks to the different refractive index of tumors compared to that of normal tissues [83].

#### 3.2.3. Light-Sensitive Smart Polymers

Materials sensitive to harmless electromagnetic radiation (mainly UV, visible and near infrared radiation), could be used as drug-delivery systems. Such materials, sensitive to light, trigger an irreversible change that causes the release of the entire drug dose, others are able to undergo reversible structural changes and behave as multi-switchable carriers, releasing the drug in a pulsating manner [79]. These light-sensitive systems have the advantage of delivering drugs to a specific site independent of the conditions of the biological environment [92]. The study of the photo-regulation mechanism of these systems and the development of new biocompatible materials for in vivo applications in drug administration [93]. Electromagnetic radiation, in the wavelength range between 2500–380 nm, applied in vivo can activate and deactivate the release of the biomolecule or drug in a specific organ or tissue, allowing excellent control of the release and reducing damage to adjacent sites [94]. UV light acts as a trigger for topical treatments [95], as the radiation used (λ < 700 nm) undergoes absorption by endogenous biomolecules and is therefore unable to penetrate more than 10 mm deep into the tissue [96]. UV treatment is, therefore, limited to therapies on the superficial layers of the skin or some internal organs. To obtain a slightly deeper tissue penetration, near infrared (NIR) light was used in the wavelength range from 650 to 900 nm, in this area of the spectrum the endogenous molecules have a minimal absorption of light, reducing the interference with the tissue.

NIR imaging techniques are non-invasive in vivo methods that allow to visualize physiological and metabolic processes [97]. Light-induced therapies fall into two categories, photodynamic therapy in which light stimulates apoptosis or necrosis by administering a photosensitizer that reacts with the light and oxygen present in the tissue to produce singlet oxygen. The second approach is photo-polymerization which induces the *in situ* formation of filling materials. (33) used for the preparation of dental composites or implants that take the shape of the implant and are applied without the use of invasive methods. Some light-sensitive systems that use azobenzene groups and similar aromatic substances are listed among the toxic compounds by the Food and Drug Administration (FDA), which limits their clinical use and pushes researchers to study alternative biocompatible materials [79].

#### 3.2.4. Biomolecule-Sensitive Smart Polymers

Polymers that respond to biomolecules can provide high specificity, higher than those that respond to physical or chemical stimuli, therefore they are being studied in the context of drug-delivery systems. An example is glucose-sensitive polymers, which use phenylboronic acid, glucose oxides or concanavaline A for the treatment of diabetes by administering insulin, which is regulated by a closed-loop feedback system, but the use of proteins such as glucose oxides or concanavaline has limitations. These proteins are difficult to immobilize, so they cause an uncontrolled release of host proteins. Furthermore, several monosaccharides could compete for glucose-binding sites. Glutathione is another polypeptide that regulates cellular redox state [80]. For example, a hybrid hydrogel has been designed, integrating genetically modified calmodulin, which can change swelling based on the response of calcium ions and phenothianzines. These systems have the potential to be used in microfluidics and miniaturized drug-delivery systems [98]. Table 1 summarizes all the smart polymer platforms discussed in this paragraph.

## 4. Flame-Retardant Coatings

Among the various textile finishings, flame-retardant treatment is crucial as it directly relates to human health and hazards. At the beginning of the 21st century, the Earth’s population annually experience a reported 7,000,000–8,000,000 fires with 500,000–800,000 fire injuries and 70,000–80,000 fire deaths [99]. Although sometimes it is not easy to discriminate whether a fire is the consequence of technical failure or human behaviour, statistics demonstrate that the presence of upholstered furniture and textiles is common in a domestic fire. According to the U.S. Consumer Product Safety Commission [99], fires involving residential upholstered furniture are the main reason for fire deaths in U.S. households, claiming 300 lives each year, causing 500 serious burn injuries and resulting in 1.6 billion dollars damages. For all the above reasons, most scientific community efforts have been addressed to investigate the flammability of textiles, fibres and fabrics [100].

Wool and silk as well as cotton and flax, as protein- and cellulose- based natural fibres respectively, are mostly used in apparels and home furnishings due to their unique advantages, including biocompatibility, biodegradability, and hydrophilicity [101]. With the increasing demand for high-performance materials, to be used besides apparel and home-furnishing applications, also for the technical textiles market [102,103,104], thermoplastic synthetic fibres are also extensively used. Polyester, polyamide, acrylic, and even to some extent polypropylene, have dominated the fibre market since the mid-1990s when they overtook cotton volumes [49]. Among other criteria, a specific classification for textile materials distinguishes them according to their fire behaviour as melting or non-melting. For natural or artificial fibres, the heated material does not melt, but the breaking of a number of covalent bonds, with subsequent destruction of the structure and the formation of volatile products and carbonaceous residues occur. Even if the limiting oxygen index (LOI) test does not provide a reliable indication of material performance in real fires, obtained values appear to be very sensitive to the composition of the material and therefore can be used to show the fire behaviour of polymers. LOI represents the minimum oxygen concentration that supports combustion of a material [47]. Since air consists of about 21% oxygen by volume, any material with an LOI less than 21% can burn easily in air. As a consequence, a material with an LOI greater than 21% can reduce the flame after removal of igniting source. A lot of research has shown that materials burning in the region between 21% and 28% can be referred to as slow-burning materials, while materials with an LOI > 28% are generally flame retardant [105]. As the majority of the natural and man-made fibres has LOI values < 25, there is a need for imparting flame-retardant finishes to make them suitable for various end applications requiring fire resistance.

Given their high flammability, if exposed to an irradiative heat flux or a flame without adequate protection, textile materials burn very easily, significantly limiting their use in those sectors where fire-proofing is mandatory. In addition, during the combustion there is the generation of smoke that can further cause severe issues and restrictions in the use because of its toxicity.

In order to overcome this issue, in the last 100 years, different flame-retardant finishes (FRs) have been developed with the aim of inhibiting or at least slowing down the propagation of a flame, were proposed [106,107]. Nowadays, some of these products are on the market with a high effectiveness according to the specific chemical and/or physical mechanisms that take place when FRs activate [108].

The key role in determining the way by which the FRs can be active in the gas or condensed phase is played by the chemical composition and the relative thermal and fire behaviour of the flame retardant molecules [108]. A fire-retardant product can contrast the combustion process, possibly stopping it, through mechanisms carried out in:

(a) condensed phase, able to influence the substrate pyrolysis by the development of non-combustible gases and the formation of intumescent which reduce the contact with the oxygen;

(b) gas phase, working on the volatile fuel products and their interaction with oxygen, by introducing inhibiting factors (free radicals).

To make a compound resistant to flame it is necessary to understand in detail the mechanism of combustion, which occurs when, by supplying heat to a substance (solid or liquid), the energy required for evaporation or pyrolysis is reached and this, in the gas phase, combining with oxygen, it creates exothermic reactions that allow the maintenance of combustion itself [109]. There are essentially three ways to avoid this [110]:
use of additives that release radicals [111];use of substances which give such endothermic reactions as to subtract all the heat necessary for the propagation of the reaction [112,113];intumescent substances (i.e., substances which after combustion form a protective solid layer, mostly carbonaceous, which prevents the exchange of energy and matter between the condensed phase and the gas phase) [114,115].

The whole polymer combustion process is summarized in Figure 6:

As happens in most cases, we try to make products in which the “active substances against fire” (flame retardants) are inserted directly into the polymers in the preparation phase, otherwise subsequently as coatings or fillers that possibly also possess more than one of the aforementioned qualities: there are substances that release radicals by preventing the propagation of reactions and at the same time form a carbonaceous barrier that protects the underlying material by preventing its combustion. For example, during combustion, polymers binding pendants can also act as FRs because the energy needed for pendant removing and cross-linking hinder combustion. In fact, the fragmentation of polymer leads, starting from cross-linking, to the charring in four stages: (i) cross-linking, (ii) aromatization, (iii) condensation of aromatics and (iv) graphitization. In this way a barrier can be built that prevents mass and energy exchange between the polymer surface and the gas phase in which radical reaction spreads up. This char wall avoids the pyrolysis of the combustible not only creating a physical medium that cannot be passed through from the fragments of polymer but isolate the matter from the heat generated reflecting it outwards.

FR are more useful when directly mixed in the polymerization phase (covalently linked to the polymer) compared to fillers that need higher concentrations to obtain the same results; moreover, in this way, mechanical properties of the polymer are better maintained. Despite this, additives are more generally used because of their wider applicability and lower costs.

It is very important to optimally design an FR and consider his mechanism of action both from physical and chemical point of view. Regarding FRs physical mechanisms, we can consider the following:
promote endothermic reactions;generate inert gases that dilute the concentration of the atmospheric oxygen;form a protecting impermeable coating.

Chemical mechanisms are mostly based on the inhibition of the oxidation reactions (gas phase) through trapping the free radical species involved in (H^•^ and OH^•^). Despite the great number of radical reactions involved in the gas phase combustion, only the following two reactions are responsible for the high process speed and the energy released:H^•^ + O_2_ → OH^•^ + O^•^ (propagation stage)(1)
OH^•^ + CO → CO_2_ + H^•^ (exothermic reaction)(2)

There are three main families of flame-retardant chemicals [100]:Inorganic flame retardants;Halogenated flame retardants;Phosphorus-based flame retardants, along with nitrogenous compounds due to their synergistic effect.

Several inorganic compounds are employed as flame retardants in fire retardant systems. Due to their characteristics, these chemicals have to be used in large concentrations or combined with other types of flame retardants to achieve desired results. For example, when applied alone, antimony oxide does not show specific flame-retardant properties. Nonetheless, when combined with halogen-based flame retardants, it acts as a catalyst causing the bromine or chlorine to break down even faster. The result of this synergistic action involves the faster release of active halogen atoms able to remove the high-energy radicals that feed the gas phase of the fire. Other inorganic flame retardants can be used independently, including aluminium and magnesium hydroxides. Both through the release of inert gases (like water vapour) and energy absorption mechanisms, these compounds interfere with the burning process creating a protective char layer or reducing the fire’s energy, respectively. In the past, halogens based polymers were a good choice because they maintained, and in some case improved, the mechanical properties of the treated materials, optimally performing this task (radical trapping). In fact, halogen-based polymers, in the flame, provide radicals that are less reactive than H^•^ and OH^•^ hampering flame propagation. Halogens have a great heat capacity and work as a “heat sink” not providing the amount of energy necessary to the pyrolysis of the polymer and excluding oxygen from the surface of the polymer.

Although the halogen-based compounds, from a technical point of view, are the most efficient FRs, some of these chemicals (namely pentabromodiphenyl ether, polychlorinated biphenyls, and decabromodiphenyl ether) were banned since they have been proven to be persistent, bioaccumulative, and environmentally toxic for humans and animals [116].

Researchers have so moved towards other classes of compounds not containing halogens. Phosphorus is an excellent candidate to substitute halogen-based polymers [117]. Phosphorus, as an FR, is highly versatile and covers a wide range of uses. Depending on its oxidation state and structure, it can be added as a filler or directly incorporated into polymer chains through homopolymerization or copolymerization [118]. Phosphorus exhibits (in the gas phase) free-radical trapping properties and acts as heat sink inhibiting complete oxidation of C to CO_2_ decreasing the energy released by combustion. Heat capacity, heat of vaporization and endothermic dissociation of P-based compounds occurring in the gas phase concur with the flame retardancy [119,120,121,122]. The most commercially used inorganic P-based compounds as FRs fillers are ammonium polyphosphate, phosphinates, phosphonates and phosphate esters; they act as intumescent forming a barrier between condensed phase and gas phase, protecting the former. Another class of P-based, phosphazenes, have recently attracted researcher interest; they are synthesized as linear or aromatic derivatives, and show interesting properties as FRs because synergistically combine the properties already mentioned of the P-based compounds with those of the N-based compounds which are described below [123,124,125,126,127].

FR treatments containing nitrogen arouse great interest because, unlike phosphorus, nitrogen is not an element among those considered to be depleted and makes the material more suitable for recycling. Among the FR compounds, those that are nitrogen-based are particularly environmentally friendly. N-containing FRs are very useful mainly in addition to polyamides and lesser extent with polyolefins (polyurethanes), but usually they are not much effective with another polymer alone [128,129,130,131]. One of the most used N-based compounds is melamine [132]: it is a very stable crystalline product (mp = 343 °C, 67% of N m/m) obtained from urea. Melamine sublimates at around 350 °C absorbing a great amount of heat and decreasing temperature of the surface of the matter exposed to fire. Under heating between 250 and 400°C, melamine decomposes in ammonia and forms cyclic compounds called melem, melam or melon, which constitute the surface layer of char [100].

In the condensed phase, nitrogen-based compounds promote the char formation. These are relatively stable at high temperatures and physically inhibit the decomposition of materials, avoiding the formation of flammable gas. Moreover, in the gas phase the nitrogen obtained by ammonia released at the pyrolysis temperature, dilute oxidizing and flammable gases. Although numerous studies referring to the synergism between nitrogen (N) and phosphorous (P) for flame retardant cotton treatments have been published so far, only qualitative observation of this phenomenon has been undertaken. Indeed, some studies [133,134] have demonstrated that it is possible to identify a real synergism between two species only through the calculation of a synergism effectiveness parameter P-N. In some cases, the two species’ effect can be merely additive or even antagonist [135].

Another class of FRs are intumescent products. Applied to materials, they are able to expand upon heating, forming an insulating fire-resistant layer at the treated surface, which protects it from further pyrolysis and burning. The result is a char residue with a characteristic foamy appearance [136,137]. Usually, intumescent systems consist of three components: (i) an acid source (e.g., ammonium polyphosphate, melamine polyphosphate), (ii) carbonization compounds (e.g., pentaerythrithol, phenol-formaldehyde resins, starch) and (iii) blowing agents (e.g., melamine, guanidine, urea, chlorinated paraffin) [138]. During combustion, first (at lower temperature), acid sources decompose providing acids (inorganic acids overall) that react with carbonization compounds (generally polyols) dehydrating them and, raising the temperature, blowing agents allow the growing of char. The final volume of the char depends basically on the amount of developing gas and on the rheological properties of the forming char. In fact, the more the char shows plastic characteristics, the more it will have considerable volume, hindering much better heat and mass transfer between condensed and gas phases. Intumescent systems result in efficient fire protection, but the efficiency is related to their thermal stability and amount. To increase the performances of intumescent is possible to use synergists, both organics and inorganics [139].

In the field of nanotechnology applied to improve the flame retardancy of materials, two approaches have exhibited the most interesting results [140,141]: the addition of nanosized particles in traditional back-coatings, and the deposition of nanosized films.

For example, clays are utilized as nanofillers, enhancing chemical (forming aluminophosphate ceramic structures) and physical (superior char strength, better thermophysical characteristics) properties.

Also, minerals, metal hydroxides, borates and hydroxycarbonates can be used as FRs [142,143]. During combustion, the metal hydroxides act as a heat sink by releasing H_2_O, by evaporation, in the same temperature range as the polymer pyrolysis. Moreover, water vapors dilute combustible gases caused by polymer decomposition, leaving on the surface a nonflammable inorganic layer. The most used mineral FRs are Al(OH)_3_ (ATH) and Mg(OH)_2_ (MDH); endothermic decomposition of i) Al(OH)_3_ occurs at 180–200 °C absorbing ± 1 kJg^−1^ of heat energy and of ii) Mg(OH)_2_ occurs at ~360 °C absorbing ± 1,3 Kjg^−1^ of heat energy. For a good FR behavior, ATH and MDH must be at least 60% of the total mass, but this negatively affects the polymer’s mechanical properties. Better performances as FR have been reached from MDH when used as nanoparticles. Zinc borates in addition to inorganic additives, are used in combustion hindering; so (x)ZnO ● (y)B_2_O_3_ ● (z)H_2_O acts as: intumescents; heat-sink absorbing, 0,5 kJg^−1^ for its endothermic decomposition developing water and boric acid; forming a boron oxide layer that protects and isolates the surface of the matter [123]. Hydroxycarbonates are a family of inorganic FRs, specifically magnesium hydroxycarbonate known as hydroximagnesite (3MgCO_3_ ● 3Mg(OH)_2_ ● 3H_2_O). This mixture of carbonate and hydroxide of Mg absorbs heat developing CO_2_ and H_2_O in a wide range of temperatures. In nature it can be found in mixed deposits with huntite (Mg_3_Ca(CO_3_)_4_) [143] that shows higher thermal stability than the hydroxides, it is often used with the latter especially when the processing temperature of the polymer could decompose metal hydroxides.

Silicon-based compounds substantially reduce the flammability of polymers. They are employed as co- and stand-alone additives, also in a nanoscaled arrangement. When exposed to heat, silicon-based compounds decompose leaving an inorganic silica residue on the surface of the polymer, forming a barrier against mass and heat passages [144]. The most commonly used silicon-based FRs are polyorganosiloxanes, specifically polydimethylsiloxanes (PDMS). The choice falls on this category of compounds because of the Si–O–Si ether bonds; they are in fact enough strong and thermally resistant, but also flexibles, assuring good FR behavior and maintaining good mechanical properties [145,146]. Often silicon-based polymers are not able to reach good performances to satisfy sufficient FR qualities, so they can be used in addition to other substances such as kaolin, montmorillonite, talc, silica, mica, and other inorganic fillers that are added with. A better approach is to use polyborosiloxanes that provide a more cohesive ceramic layer to the surface, thus offering better protection against fire [147].

These act overall as physical barrier avoiding heat transfer to the surface of polymer and hence preventing pyrolysis. Si-based nanomaterials (clays), besides providing a protective layer, act as catalysts promoting char formation. The graphite char forming during combustion, and its effectiveness as a FR, oriented researchers to test carbon nanotubes on this application. Indeed, it appears that carbon nanotubes act as a mirror that reflects much of the electromagnetic radiation that accompanies combustion, protecting the underlying layers.

Due to the electrostatic attraction between positively and negatively charged layers on the surface’s substrate in layer by layer (LbL) assemblies, [148] the resulting interactions are robust and independent from the substrate size and topology. Among several investigations reported in the literature, Carosio et al. [149] exploited the LbL technique to improve the flame resistance and to solve the dripping of polyethylene terephthalate fabric. Tests, carried out according to the ASTM D6413 standard [150], showed that the treatment reduced the burning time by 95 % and eliminated melt dripping phenomena.

Furthermore, in the last few years, the sol–gel technique has been exploited in order to protect polymer surfaces with nanosized films exerting a thermal shielding effect, thus improving the flame retardancy of the treated materials [135]. Generally speaking, the investigated coatings can be divided into two groups, namely: fully inorganic, and hybrid organic–inorganic systems. Concerning the sol–gel based fully inorganic coatings on cotton, the effect of the type of hydrolysable groups on the overall flame retardant features of cotton samples was investigated [151].

In particular, tetramethyl orthosilicate (TMOS), Tetraethyl orthosilicate (TEOS) and tetrabutylorthosilicate (TBOS) were utilized to develop silica coatings on cotton samples [152]. The findings observed by flame spread tests, realized in vertical configuration, confirmed a well-defined effect of the type and chain length of the alkoxy group in the precursor on the residue at the end of the tests. The longer the chain length of the precursor, the worse the overall fire performance.

Furthermore, when an organic component is combined into a sol–gel solution, or incorporated in hydrolyzed metal alkoxide precursors, the obtained finishes convey new fire-retardant properties to treated polymer surfaces. Indeed, in this composition, the hybrid organic-inorganic coatings show the properties of both phases or even new ones [153]. Therefore, the thermal stability at high temperatures proper of inorganic constituents can be combined, for instance, with the flame retardancy of organic phosphorus-based compounds. The concurrent presence of P and Si elements can be exploited for preparing hybrid organic–inorganic coatings that behave, at the same time, like char promoters (because of the phosphoric-acid source) and thermal shields (due to the inorganic component).

Among the most recent results obtained [108,153], some papers showed high flame retardant properties in treated cellulosic-based materials.

The combination of TEOS, sodium alginate and phytic acid resulted in a coating able to enhance the thermal stability of cotton, also suppressing smoke emissions [154]. Moreover, self-extinguishing cotton fabrics thanks to the deposition of a sol–gel precursor containing 3-glycidyloxypropyl trimethoxysilane modified with *N-*(phosphonomethyl) iminodiacetic acid, hydrolyzed and condensed in the presence of TEOS, were obtained [130]. The coating decreased the heat release rate, the peak of heat release rate, and total heat release values. At the same time, findings showed an increased time to ignition and a residue higher than 26% at the end of the combustion tests. Furthermore, flammability tests performed on fabric samples finished with 2,2-dimethyl-1,3-propanediol-(3′-triethoxysilanepropyl)phosphoramidite, a P-containing siloxane product, showed a considerable increase of LOI (30%) [155]. Treated fabrics exhibited high char-forming character, achieving self-extinction without afterflame or afterglow.

## 5. Industrial Coatings

Industrial coatings play a key role in modern society, contributing significantly to our care, health and wellbeing. The protection provides by sealants, paints and other coatings preserves and extends the lifetime of buildings, transportation vehicles, industrial equipment and other infrastructure [156]. Many of the actual coatings for corrosion protection represent a threat to the environment. Therefore, the main strategy is the development of innovative materials with better performance that are eco-friendly at the same time.

In recent years, the world of materials engineering was involved in developing new functional coatings to modify surfaces’ properties through the sol–gel process. Silica–based nanostructured coatings are, indeed, one of the most advanced techniques employed in nanotechnologies [157]. This type of method, which leads to the formation of organic–inorganic hybrid coatings with a nanocomposite structure, can modify some properties of the materials under study, such as the resistance to abrasion, wettability, impart antibacterial characteristics and resistance to UV radiation, in order to release bioactive substances [158]. Over time, the advantages of organic-inorganic hybrid materials were proved by several researchers, which established that these compounds are obtained from the structural incorporation of organic groups, such as, vinyl, acrylic, epoxy, amine etc., in the sol–gel precursors through Si–C bonds [158,159]. In detail, the inorganic part is formed by alkoxides of silicon or other transition metals through a hydrolysis reaction, whilst the organic one is constituted by molecules, which can differ in their structure from the length and branching of the alkyl chains [160,161].

The inorganic moiety gives to the hybrid material some properties among which, hardness, chemical resistance and adhesion to the substrate, whilst the organic one provides elasticity, toughness and low curing temperatures [159,162]. Curing time and temperature, pH, aging concentration, molar ratio and nature of the precursors are all factors that contribute to the final chemical structure of the silane produced [159,161,163,164].

In literature, there are several works which explain different types of coatings and their applications as anticorrosion, anti-scratch, antiadhesive, antimicrobial, antifouling and self-cleaning coatings [165,166,167,168]. However, one of the most important research fields for these hybrid materials is the application of sol–gel coatings on metal surfaces or composite with the aim to work as a protective barrier against corrosive species or as a decorative agent, instead of chromates protective pretreatments, considered toxic and carcinogenic [169,170]. Moreover, the treatment of sol–gel hybrid coatings by doping with environmentally friendly inhibitory materials is a promising approach that prevents the corrosion of the substrate by improving its mechanical properties [171,172,173]. The hybrid material used as a coating for corrosion mitigation need to show resistance to the abrasion processes and good adhesion to the substrate surface, including copper [174], magnesium [167,175] and aluminium based alloys [176,177], stainless [178], galvanized [179] and carbon steel [180]. Generally, the organic–inorganic hybrid compounds, which are featured by covalent bonds between the organic and inorganic components (class II), show properties that are well suited to an anticorrosive product, such as good corrosion protection, good scratch resistance, good corrosion protection, high hydrophobicity and low dielectric constants [161,181]. In this paper, recent studies have been reviewed to show how sol−gel surface functionalization with silica-based nanosols represents an important technology for a variety of applications in industrial coatings [157]. Table 2 summarized composite coatings features used to prevent corrosion in steel and metallic materials reported in the literature of the last 10 years.

Figuera et al., show the application of siloxane functionalized coatings on different metallic substrates and assert that most of them were used on aluminium and iron-based alloys [160,179,194]. The influence of the molar ratio of two compounds, generally (3-glycidyloxypropyl)trimethoxysilane (GPTMS) and tetraethylorthosilane (TEOS), on the structure of the molecular net produced by sol–gel technology for the protection corrosion coating were the main objectives of many investigations, as reported by Alcantara et al. [159]. Kirtaky et al. and Aparacio et al., show a similar study employing glycidoxypropyltrimethoxysilane (GLYMO) and aminopropylethoxysilane (AMEO) for the first one and a mono-substituted methyltriethoxysilane (MTES) with dimethyldiethoxysilane (DMDES) for the second one as precursor materials [189,193]. Over time, the sol–gel coating techniques have evolved with the implementation of synthetic or natural inhibiting agents into the hybrid matrix. In this regard, Balan et al. claimed that nanoparticles added in the silane sol–gel networks can improve corrosion resistance. They evaluated the electrochemical corrosion resistance of low carbon steel coated with hybrid organic-inorganic sol–gel film, made by 3-glycidoxy-propyl-trimethoxy-silane (GPTMS) and tetra-ethyl-ortho-silicate (TEOS) precursors and filled with silica or alumina nanoparticles [180]. A similar study was performed by Vivar Mora et al., which assessed the impact of silica nanoparticles on the morphology and mechanical properties of a GPTMS/TEOS sol–gel coating [186]. The authors observed that features, such as hardness, elastic modulus, brittleness and fracture toughness, were improved when the mild steel surface was treated with silica NPs, leading to a more durable and corrosion resistant coating. The influence of nanoparticles in the synthesis of industrial coating was also evaluated by Rivero et al., who doped layers of a sol–gel hybrid matrix (GPTMS/ MTES) with TiO_2_ NPs deposited on flat samples of aluminium alloy [176]. Hybrid silica sol–gel coatings (GPTMS/TEOS) doped with cerium nitrate were synthesized and characterized by Tringer et al. [188]. The authors revealed that Ce(NO_3_)_3_·6H_2_O encapsulated into the sol–gel matrix together with silica particles affect the corrosion properties of the coating improving the material resistance (Aluminium alloy). The enhancement of this feature was attributed to the combination of the silica matrix hindering properties with the active protection of Ce(NO_3_)_3_. In order to obtain coatings with better anticorrosive properties, Adsul et al., investigated the effect of Ce^3+^/Zr^4+^ corrosion inhibitors loaded into halloysite nanotubes dispersed in a GPTMS/TEOS sol–gel matrix on the corrosion resistance. The studies were conducted on magnesium alloy substrates and demonstrated how these cationic inhibitors prolonged corrosion protection [167].

The evaluation of anticorrosive properties of hybrid sol–gel coatings hosting graphene oxide as corrosion inhibitor was presented by Maetzu et al., which demonstrated the protective power enhancement when these formulations are applied to aluminium alloys [191]. The use of graphene oxide can be exploited for the design and the application of novel multi-layered coatings in the engineering fields. However, silica or aluminium nanoparticles, as well as the other doping agents mentioned above, are not the only inhibitor agents that contribute to favouring the protective action of sol–gel matrix.

Dalmoro et al. examined the effects of the ethylenediamine tetra(methylene phosphonic acid) (EDTPO) addition to a hybrid sol–gel matrix prepared with vinyltrimethoxysilane (VTMS) and tetraethylorthosilicate (TEOS) [192], whilst Peerin et al., studied the influence of triethylenetetramine (TETA) in GPTMS/TEOS sol–gel formulations [182]. In the first case, the results evidenced the formation of Si-O-Al and P-O-Al bonds in VTMS/TEOS/EDTPO system, which were considered responsible for the good anticorrosive behaviour. In the second case, the presence of TETA in the sol–gel matrix led to the formation of interconnected epoxy-amine and silica networks, acting as a catalyst for the condensation of GPTMS and TEOS, as well as improving barrier proprieties. Moreover, considering the research works in which organic compound was employed for the preparation of industrial anticorrosive coatings, Agustín-Sáenz et al. proposed the incorporation of mono-phenol (AP) and bi-phenol (BPA) organic precursors in an epoxide functionalised-silica-zirconia matrix, in order to obtain a sol–gel coating material with higher corrosion protection for zinc-coated steel [187]. In this work, sol–gel matrix containing AP and BPA showed improved corrosion protection properties owing to the formation of a highly cross-linked network due to the organic compounds, which acted as a bridge between the epoxide functions of two GPTMS molecules.

Duran et al., reported a review in which the structure, properties and features of coatings produced in the first decade of the 2000s were discussed. They assert that good oxidation resistance was achieved using mixed or hybrid organic–inorganic SiO_2_ layers. In particular, the authors referred to alkylalkoxysilanes linked to polymerisable groups for the coating synthesis and the alternative use of a doping agent, such as an environmentally friendly inhibitor, to improve the corrosion resistance [171]. In this regard, a novel practice to replace high volatile organic content coatings consisting of toxic agents, such as hexavalent chromium compounds, was the use of water-based coatings modified with green active agents. Izardi et al., as well as Hamidon et al., proposed sol–sol–gel formulations treated with aqueous–based basil extract and caffeine, respectively [183,185]. The results obtained by Izardi et al., confirmed the precipitation of basil-containing thin film on the mild steel specimens, which led to an improvement of corrosion processes. In addition, Hamidon et al., asserted that caffeine-doped silane coating offered a high corrosion resistance for mild steel samples.

Finally, some researcher, such as Abdelaziz et al. and Calabrese et al., studied new anticorrosion coatings, exploiting the high chemical affinity of a zeolite filler with the silane matrix [184,190]. Nowadays, the research on innovative materials has also focused on developing sol–gel based coatings for the construction, finishing and decoration of structural materials different from steel, such as Portland cement.

In this regard, the sol–gel technique represents a possible method to improve the high- temperature resistance on cement composite. As is well known, cement is an inorganic binder, which is featured by high strength and fast hardening, but if it is subjected to high temperatures or some of other external agents, such as an acid attack, its structure undergoes several cracks and even bursts in extreme conditions, with it losing its structural role quickly. The most commonly reinforced material is steel fibre, but it easily undergoes deterioration over time [195]. In light of these considerations, Xu et al., show an improvement of the resistance to high temperatures when Portland cement is filled with a carbon fibre felt and treated with an Al_2_O_3_ sol–gel coating [196]. Moreover, it is possible to treat cementitious material with functional TiO_2_ coatings, as reported by Costantino et al. [197]. The authors evaluated the photo-activity the hydrophilic properties of the material under study, before and after the TiO_2_ thin film deposition.

Further investigations on new corrosion inhibitors encapsulated in a sol–gel matrix should be performed. Smart coatings with self-healing properties are the possible materials of the future in this field. The self-healing properties of an organic–inorganic hybrid coating achieved with the incorporation of additive agents, lead to the reinstatement of the material structure and restore the protective coating wholeness due to external factors (e.g., pH, UV, the presence of Cl), by avoiding the initiation of the corrosive process.

## 6. Nano/Ultrafiltration Membrane Coatings

Membranes for nanofiltration (NF) and ultrafiltration (UF) processes in the water treatment field are employed to remove the most common organic and inorganic contaminants (e.g., natural organic matter, pharmaceuticals, inorganic salts, organic dyes) with high efficiencies, through different removal mechanisms like electrostatic repulsion, size/steric exclusion, hydrophobic adsorption etc. [198,199,200,201].

The major problem of this technology, mostly in low-pressure processes, is the membrane fouling, that negatively affects membrane performance. In recent years, a proposed solution to reduce the membrane fouling and improve their durability, selectivity, retention and permeate flux, consists in the surface modification of membranes. This can be easily done using coatings that represent the most efficient approach, because of its easy processability, which involves chemical modifications to change surface properties of membranes. In particular, membrane coatings based on nano-sized materials like graphene oxides (GOs), carbon nanotubes (MWCNT), and titanium dioxide (TiO_2_), thanks to their higher hydrophilicity and the capacity to reduce pore size and increase charge effects of membranes, represents a technology of growing interest in the water treatment field.

GO nanosheet coatings for membranes have good chemical stability, exceptional transport properties, and excellent mechanical stiffness and strength; in particular, the nanochannels of GO sheets 1 nm wide, by a sieving mechanism, can reject larger molecules when the water passes through the membrane filter [202,203,204].

Some studies have reported the removal of several contaminants by a GO membrane in combination with polymeric membranes like polyvinylidene fluoride (PVDF), polysulfone (PS), polyamide (PA), or poly(ether sulfone) (PES), fabricated with various methods in which GO layers are bonded or unbonded together (Figure 7).

GO membranes supported on microporous substrates of PVDF exhibit high rates of rejection for organic dyes methylene blue and rhodamin WT, under a transmembrane pressure of 50 psi (0.34 MPa), in a range of 46–66% and 93–95%, respectively and NaCl and Na_2_SO_4_ in a range of 6–19% and 26–46% respectively, depending on the specific number of GO layers deposited [205]. Also ceramic membrane coated with GO, at the expense of a low water permeability compared with the pristine membrane, show retention efficiencies of NOM (natural organic matter, humic acid and tannic acid), pharmaceuticals (ibuprofen and sulfamethoxazole), and inorganic salts (NaCl, Na_2_SO_4_, CaCl_2_, and CaSO_4_), in the order of 93.5%, 51.0%, and 31.4%, respectively, much higher than the pristine membrane [206]. Moreover, the abundant oxygen-containing functional groups feature of GO (carboxyl, carbonyl, hydroxyl and epoxy, distributed at edges and structural defects of GO flakes), which causes a high negative charge on the surface of the membrane, enhancing the hydrophilicity, retention, and antifouling properties. Multiple layers of GO coated on PES ultrafiltration membrane obtained via vacuum filtration of a GO suspension, exhibit a NOM rejection in the order of 31–67%, based on the number of GO layers coated, and a water flux change of the permeate less than ±10% [207]. GO-coated UF membranes, thanks to their underwater superoleophobicity and low oil-adhesion, are very effective in oil-in-water emulsion separation and the removing of oil droplets with sizes in the micrometer range, to obtain water with low oil/grease concentration. Porous polyamide (PA), with 200-nm 3D pores, coated with GO by vacuum filtration for antifouling oil/water separation, show an excellent antifouling performance thanks to the low oil adhesion on the membrane surface, resulting from the optimized micro-/nano-hierarchical roughness of the GO in particular with a 10 nm thickness and exhibit a 100% recovery by surface water flushing [208]. The underwater oleophobicity of GO sheets can be tuned by oxidative etching with ultraviolet (UV) light, to create or enlarge structural defects and introducing oxygen groups around them, to improve potential applications of GO coatings in oil/water separation, oil-repellent materials, microfluidic devices, anti-bioadhesion materials, and robust antifouling materials [209].

The most common fouling agents in water present a negative charge, so they are sensitive to a negatively charged membrane surface due to the electrostatic repulsion, as we have seen in the case of a GO-coated membrane. Several methods to enhance the surface charge of membranes have been developed, and one of them is the fabrication of electrically conducting membranes (ECM) to perform electrofiltration processes by the application of an electrical field. This type of membrane can be produced by coating UF membranes with conductive inorganic materials like carbon nanotubes (CNTs) or multiwalled carbon nanotubes (MWCNTs) that can also be used as inorganic fillers to fabricate nanocomposite membrane, to improve their performance also for the desalination of water [210]. CNTs and MWCNTs, before their use in the fabrication of nanocomposites or coatings for their application in membranes, are treated by acid treatment, that however can lead structural damage of MWCNTs, or by coatings with some molecules like poly(vinyl alcohol), polyaniline or polydopamine, to enhance their hydrophilicity [211]. An example is represented by a thin film made by cross-linked poly(vinyl alcohol) and carboxylated MWCNTs, deposited by pressure on a PS membrane. This system, used in an electrofiltration cell, in which a cell potential of 3–5 V and fields of 9–15 V/cm are applied, demonstrates the inhibition of a negatively charged fouling agent, represented by alginic acid, and the potential of CNT-based coatings on the reduction fouling rates [212]. Also composite membranes based on CNT-conjugated polymers (i.e., PANI, polypyrrole), fabricated through a process of electropolymerization of aniline on a CNT substrate under acidic conditions, the latter obtained by a coating of a PS membrane with a CNT suspension performed by pressure deposition, demonstrate that the application of an anodic potential to the ECM surface, is able to degrade a model organic contaminant (methylene blue) through an electro-oxidation process, also showing an electrochemical in situ membrane cleaning capacity [213]. To reduce membrane fouling by photo-oxidation, there is another solution that can be represented by the use of Titanium dioxide coatings. Photoactivity properties of the semiconductor titanium dioxide, exhibiting under UV irradiation, can be exploited for the photodegradation of smaller organic molecules entrapped in UF membranes. In particular, photoactive anatase membranes on asymmetric ceramic supports, prepared by slip-casting on asymmetric tubular supports in alumina, subsequently immersed in nanocrystalline anatase sols, show a high retention capacity of colloids and macromolecules, ensured by the separative top layer and the photodegradation of smaller organic molecules performed by UV irradiation of the opposite side of the membrane [214].

Manganese oxide and iron oxide coatings for catalytic membranes are also evaluated for the retention and removal of total organic carbon (TOC) in ultrafiltration processes, which is proven to depend on the number of times the membrane was coated with the metal oxide nanoparticles. This type of membrane is produced by coating, using a layer-by-layer self-assembly technique, of commercial UF ceramic membranes, that can also be cleaned from fouling with Distilled DeIonized (DDI) water using an ozonation-filtration. Hybrid ozonation-ceramic membrane filtration performed with Mn oxide-coated membranes, have given the best results in comparison with other metal oxide coatings (titanium oxide and iron oxide), thanks to the excellent catalytic properties of manganese in the oxidation of organic material and then in the reduction of TOC in the permeate [215].

Another application of metal coatings in filtration techniques, is their use as metal mesh coatings for application in oil/water filtration. An example is an eco-friendly iron-based, with a micro/nano-structure, metal mesh coating, produced by immersion of a stainless steel mesh in a solution in which is performed the reduction of FeCl_2_ with NaBH_4_, resulting in a membrane with underwater superoleophobicity, with an oil contact angle as high as 152°, that can separate oil/hot corrosive water mixed liquid efficiently, with a separation efficiency >96.2% [216].

A further approach of water filtration in water treatment processes, consists in the use of other types of membranes like chemically-functionalized membranes (CFMs), in which are incorporated selective ligands or ion-exchangers for the extraction specifically chemical species, bulk liquid membranes (BLMs), emulsion liquid membranes (ELMs), supported liquid membranes (SLMs) in which two aqueous phases (feed and stripping) are separated by an organic liquid as interphase and polymer inclusion membranes (PIMs). PIMs, more stable than SLMs, have acquired significant importance in recent years in the field of separations and extraction of organic molecules and metal ions (Au(III), As(V), Cd(II), Co(II)) from water [217]. This type of membrane also has important relevance in the preconcentration of antibiotics (sulfonamides and tetracyclines), which is highly influenced by sample pH, in environmental water samples [218]. PIMs are thin, flexible and stable films made by casting a solution containing an extractant (or carrier), usually an ionic liquid for the selective extraction of the target chemical species, a plasticizer to improve the elasticity of the membrane or modify the solubility of the extracted species and a base polymer such as cellulose triacetate (CTA) or poly(vinyl chloride) (PVC) [219]. The extractant agent in PIMs can also be represented by inorganic species that have a high capability of ion exchange and cation fixation like clays, in particular montmorillonite clay [220]. A PIM membrane based on PVC, whit montmorillonite and ionic liquids (Aliquat 336, Alicy and thiomalic acid) like extractant and plasticising agents, for the absorption and preconcentration in particular of Sn^2+^ from water samples, can be produced in two different methods: casting the clay modified with organosilanes (by a sol–gel method with dynasylan and APTES) in the solution of membrane (one pot method) or by coating the PIM membrane with the organoclay (layer by layer method).

The research of new technologies for the improvement of retention and regeneration properties of membranes in the waste water treatment, is constantly expanding, but strong evidence, as reported above, shows that the simplest and most efficient methodology for changing the surface properties of membranes, is represented by the use of membrane coatings based on various nanomaterials, which in particular allow easy cleaning, through simple washing or electrochemical methods, from fouling or oil residues for their reuse, in a circular economy perspective.

## 7. Antifouling (AF) Coatings

Preventing the growth of microorganism on several surfaces, namely known as biofouling, is one of the crucial application of nanocoatings. Over the years, it has been a significant challenge in the design of antifouling coatings owing to the diversity of fouling organisms and their adhesion mechanisms [221]. The term fouling generally refers to an undesirable process in which a surface becomes encrusted with material from the surrounding environment. In the case of biofouling, the material consists of organisms and their by-products. The marine biofouling process starts from the moment the surface is immersed in water and takes place in distinct steps of growth [222] as follows:Adsorption of organic molecules such as proteins, polysaccharides and glycoproteins that form a conditioning film.Primary colonization with a settlement of microorganisms such as bacteria and diatoms, by creating a biofilm matrix.Secondary colonization consisting mainly of a biofilm of multicellular species (e.g., visible algae and invertebrates) generally called microfouling.Tertiary colonization including macrofoulants which consists of shelled invertebrates like barnacles, mollusks and sponges.

The unwanted adhesion of microorganism (Figure 8) may affect the performance of devices, causing several problems for marine industries due to corrosion and hydrodynamic drag, which leads to elevated fuel consumption and, consequently, to higher maintenance costs. In view of the variety of problems biofouling poses, altering the surface properties of substrates via molecular engineering is essential. In addition, this biofilm can lead to microbially induced corrosion (MIC) due to H_2_S formed by the bacteria, especially sulfate-reducing bacteria, creating surface cracks and leading to stress corrosion and potential release of products into the seawater.

### 7.1. Tributyltin-Based Antifouling Solutions 

The search for solutions to these problems stimulated extensive research on coatings, which prevent and control biofouling known as biocidal and non-biocidal coatings. In the middle 19th century, most of the control procedures involve paints with dispersed chemical biocides, i.e., toxic compounds for the marine biological organisms. Biocide-based antifouling coatings function by slow leaching of the incorporated biocides into the coating. It is important that the biocide does not have any adverse effects on marine life while carrying out its antifouling activity. For example, historically, tributyltin (TBT) represents the easiest and cheapest biocidal agent to prevent the growth of biofilms because of its excellent antifouling efficiency and its wide-range activity, but it is no longer used due to its toxicity. Basically, the use of triorganotin derivatives seemed to be the answer to the several problems of the biofouling effects until environmental concerns with TBT started to be raised. For this reason, TBT-based AF coatings were banned worldwide in 2008 as a follow up of an international convention [223]. After the ban of tributyltin due to the harmful effects on marine organisms, copper-based paints were applied in marine antifouling applications. However, the short-time activity of the coatings and the risk of heavy metal ions leaching and bacteria resistance directed the researchers to look for alternative approaches.

### 7.2. Eco-Friendly Silica-Based AF Formulations

In this scenario, intense research started to be aimed toward the development of environmentally benign and economically AF alternatives systems able to inhibit bacterial settlement like biocides used to [37,49,50,224]. In view of the current societal expectation of using new clean, flexible and effective products with respect to the aforementioned unhealthy ones, the sol–gel coating technology is probably one of the most important platforms for the development of eco-friendly antifouling and fouling release formulations [225]. Silane-based coatings have been widely used in biofouling mitigation treatment of different surfaces, providing a non-toxic alternative to biocidal-based AF coatings [226]. Depending on the appropriate choice of the silica precursor and its related characteristics, the sol–gel technique offers the possibility to create hybrid nanocoatings within which it is possible to include substances that are able to introduce specific potentialities such as antibacterial, self-cleaning or water repellency activity [49]. In this context, the sol–gel nanotechnology can be an excellent tool to convey new properties to surfaces and to combine different functionalities in a single material, contributing for example to obtainment of low fouling surfaces thanks to the unique structure and properties of hybrid inorganic–organic silica-based coatings.

### 7.3. Nanostructured Biocide-Based AF Formulations

Application of nanotechnology has been proven to effectively improve the anti-bacterial properties and the high durability performance of the doped-silane coatings, thanks to the nanomorphology of the filler used. According to the literature, in fact, a number of antifouling products have been developed by using micro-encapsulation of inorganic biocides such as silver oxide (AgO), zinc oxide (ZnO), copper oxide (CuO), titanium dioxide (TiO_2_) and selenium. Based on different analysis, the AF effects of the metal oxide can be attributed to the reactive oxygen species (ROS) produced by photocatalysis and to the slow release of toxic metal ions when the metal oxide nanomaterials are exposed to visible light irradiation [227]. Among these, ZnO has largely been used as an antimicrobial agent since it showed the strongest and the most versatile antimicrobial activity against several marine fouling bacteria. The AF mechanism is still not very clear, but three different mechanisms of the action of ZnO are proposed: (1) the micro/nano-topography of the coatings; (2) the toxicity of Zn^2+^ ions released from the coating which binds with the bacteria, affecting the bacterial growth cycle and (3) the aforementioned production of reactive oxygen species (ROS) under an appropriate environment. ZnO nanoparticles and nanorods coatings are attractive due to an easy and controlled sol–gel process of fabrication and because of their increased stability and lower toxicity [228]. In fact, due to the slow release rate of zinc ions, no harmful effect was detected on the non-target organism. In addition, zinc is an important constituent of DNA and RNA polymerase enzymes which is essential for fish growth [229]. Titanium dioxide nanoparticles represent another common antifouling agent which is not only cost-effective, but also non-toxic to the marine organisms. Bactericidal nanocoatings can also be included for light-activated photocatalysts such as TiO_2_ or other photosensitizers activated by light irradiation with appropriate wavelength to kill the bacteria. The main mechanism is the production of reactive oxygen species (ROS) such as superoxide anion (O_2_^−^), hydroxyl radical (OH•), and phototoxic singlet oxygen (^1^O_2_) under light irradiation providing antibacterial properties [230].

### 7.4. Hydrorepellent Coatings

Due to the limitations involved in the use of biocides in antifouling marine systems, several technologies have been developed to produce new coatings aimed to interfere directly with the adhesion of microorganism as a result of topography or surface chemistry, thus demonstrating a fouling release activity (FR). Many attempts have been made by reducing the critical surface tension using fluoropolymers combined with silicones. Moreover, deepen research has also been made on the development of hydrophobic surface able to inhibits the initial step of microfouling settlements [231]. Fluoropolymers can be successfully used to form anti-adhesion surfaces by virtue of exposed CF_2_ and CF_3_ groups at the interface that can reduce the attachment of fouling, also depending on the degree of mobility of the fluorine atoms [232,233]. These fluorinated coatings, in addition to having low adhesive properties, show a good antimicrobial activity towards different kind of bacteria. Moreover, the eventual biocide effect due to the product released in the liquid medium evaluated by microbial cell experiments showed no biocidal effects [226]. However, besides the high cost of fluoropolymer-modified silicons, this non-biocidal approach is only effective for vessels cruising at relatively high speeds and that are not inactive for a prolonged period. Of course, the low efficacy of these coatings has limited their use on a larger scale. In the last few years, researchers have been trying to produce new nanostructured surfaces by simply taking inspiration from the natural antifouling behavior of some marine organisms. The surface roughness of sharks, for example, allows them to remain free from fouling and the antifouling mechanism is based on the reduced number of available attachment sites by weakening attachment strength when the surface is subjected to hydrodynamic forces [234]. Despite designing/reproducing the marine antifouling systems based on surface texture has been successful, it still needs more careful experiments to investigate. On the basis of all these considerations, it seems that a one-dimensional approach could not be enough for inhibiting the attachment and growth of all the organisms implicated in marine biofouling. In this regard, the best way to develop more promising antibacterial nanocoatings with a higher performance consists of combining different approaches in order to achieve synergistic effects, for example combining nano surface roughness together with amphiphilic or zwitterionic surface chemistry to act on the bioadhesives.

## 8. Conclusions

This review describes the synthesis of different type of nanostructured coatings and surface finishing, the related properties and the most important applications. The published results show that the potentiality of smart materials in the development of advanced, reliable and robust wearable sensors and also several innovative applications would be the result of the combination of smart coatings, textile fabrics and microelectronics.

The use of polymers combined with nanomaterials or functional nanocarriers in drug-delivery systems finds their main application in the medical sector, in particular in cancer diagnostics and therapy and in traditional oral delivery systems. The combination of different nanosystems, which exhibit different responses to external stimuli, can, in some cases, enhance the desired effect or influence each other in the desired way. Nanomaterials are also the most promising materials regarding flame retardancy, antifouling and anti-bacterial activity. Despite their nano dimensions, nanoparticles have great superficial development, providing high results with a low amount of matter. As a matter of fact, in wastewater treatment the simplest and most efficient methodology to modify the surface properties of filtering membranes, thus improving their retention and regeneration properties, involves the use of membrane coatings based on nanomaterials.

The recent literature also highlights the characteristic proprieties of sol–gel polymeric materials and their possible employment in the development of widely employed 3D matrices for the development of useful coatings for corrosion mitigation on different metallic surfaces or structural consolidation, as well as smart, fire-resistant, and antifouling silica-based coatings.

The availability of prototypes as concrete results of the multidisciplinarity in nanotechnological sectors highlights the innovation of such advanced materials in daily life, the effects on modern society in terms of costs and quality of public health, as well as the importance of even more advanced research activity in this research field. In this panorama, even if significant improvement has been provided by the more recent smart textiles, several challenges should still be addressed. Besides the fabrication cost and the lack of standards, a relevant challenge could be represented by the total integration of electronics in textile polymers for application fields like wound monitoring and healing, and surgery.

There are different scientific and technological questions, which have not been resolved yet, such as coatings nanostructure and their thermal stability conditions. Despite these questions, experiments indicate that nanostructured coatings have powerful technological potential and offer important industrial progress.

## Figures and Tables

**Figure 1 materials-14-02733-f001:**
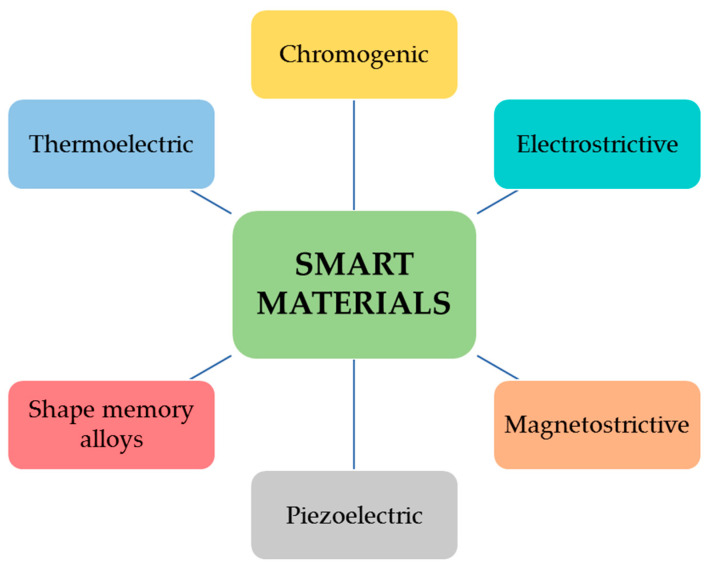
Classification of smart materials [4]. Reprinted from “Polymer Nanocomposite-Based Smart Materials”, edit by: Rachid Bouhfid, Abou el Kacem Qaiss, Mohammad Jawaid, in Woodhead Publishing Series in Composites Science and Engineering, Chapt. 1: “Introduction: different types of smart materials and their practical applications”, Pages No. 1–19, Copyright (2020), with permission from Elsevier.

**Figure 2 materials-14-02733-f002:**
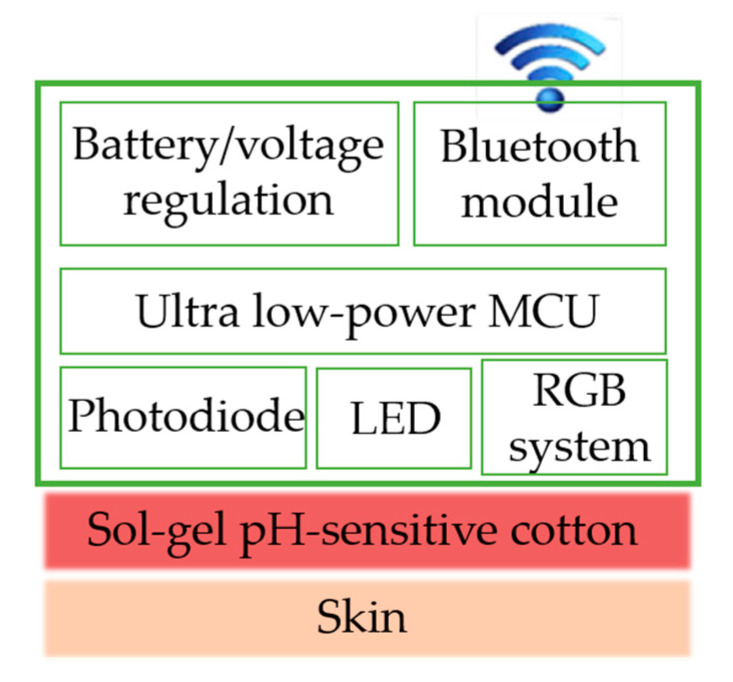
Block diagram of smart halochromic sensor.

**Figure 3 materials-14-02733-f003:**
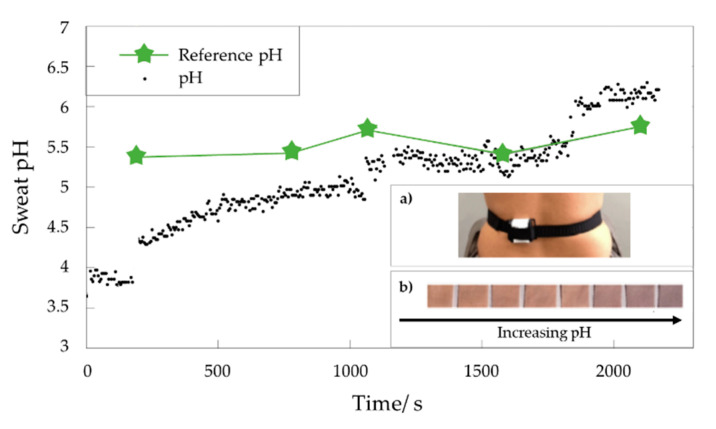
Comparison of sweat pH measurements carried out by wearable sensors and a reference pH meter as a function of time. In insets, the on-body position of the device during physical activities (**a**) and colour changing as a function of increasing pH values (**b**). Adapted with permission from [56] Copyright (2016) American Chemical Society.

**Figure 4 materials-14-02733-f004:**
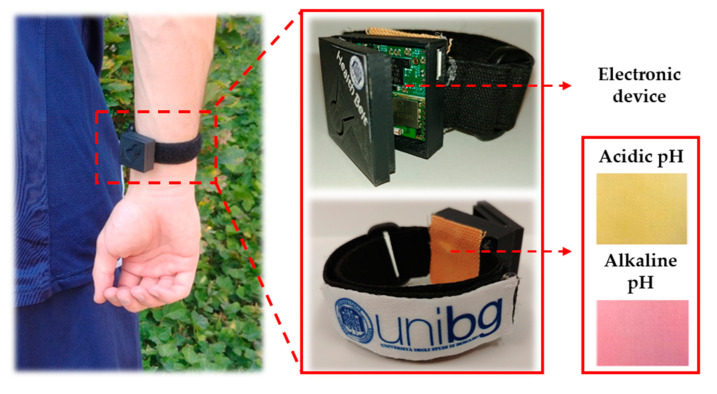
Prototype “Health Belt” for the real-time monitoring of sweat pH.

**Figure 5 materials-14-02733-f005:**
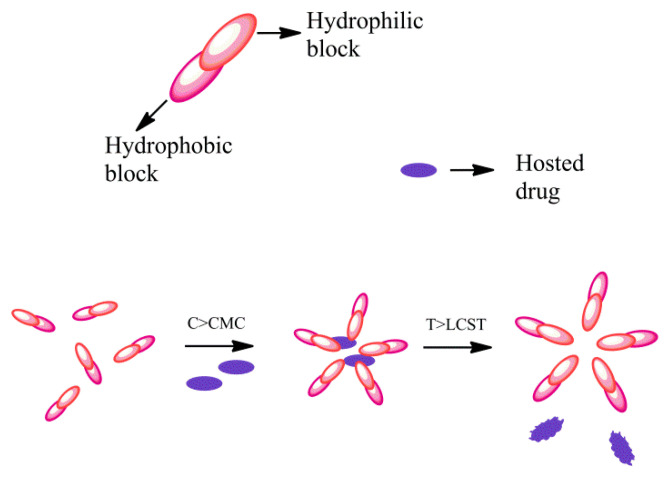
Scheme of behavior of thermo-responsive amphiphilic polymers.

**Figure 6 materials-14-02733-f006:**
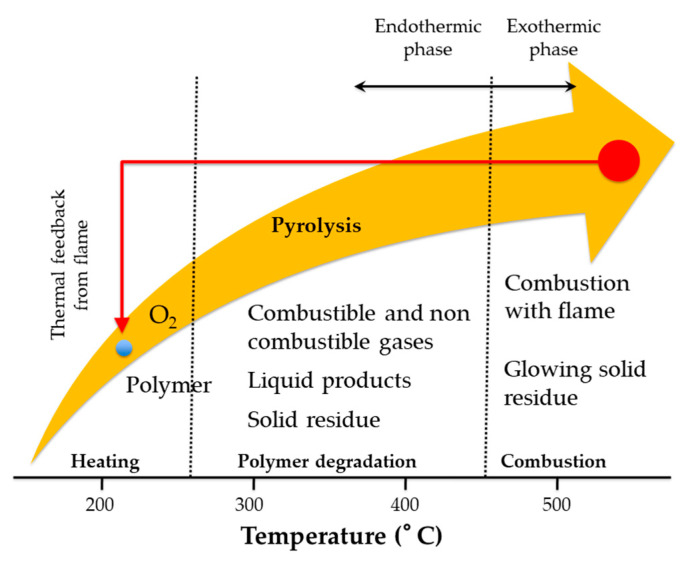
Schematic representation of polymer combustion process.

**Figure 7 materials-14-02733-f007:**
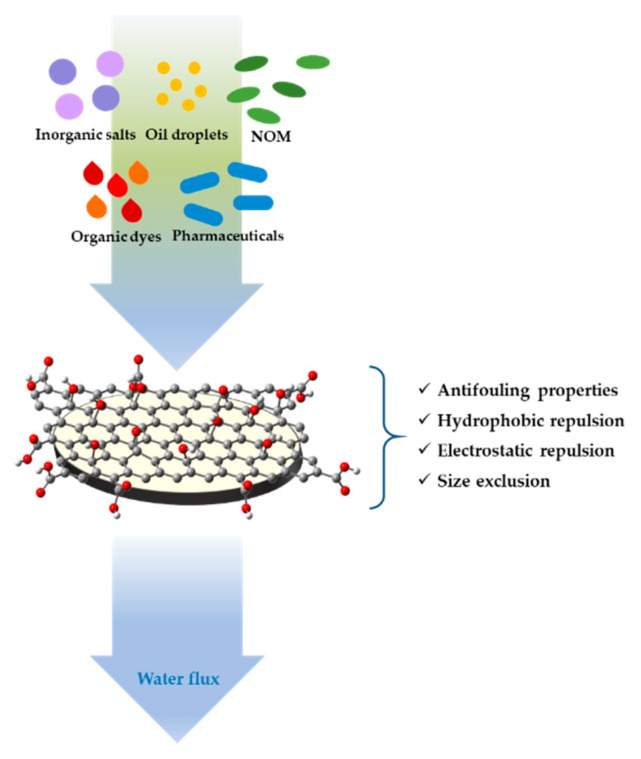
Schematic representation of water contaminants retention of graphene oxide (GO) coated ultrafiltration (UF) membrane.

**Figure 8 materials-14-02733-f008:**
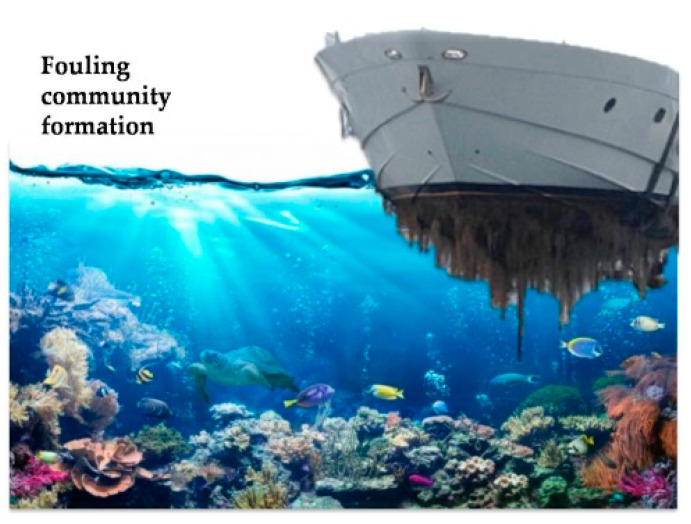
Illustration of the undesirable accumulation of marine biofouling on artificial surface immersed in seawater.

**Table 1 materials-14-02733-t001:** Advantages and disadvantages of using the various smart polymer platforms.

Smart Polymers System	Advantages	Disadvantages	References
Temperature-sensitive	Temperature is an easily controllable parameter. Most of these systems are biodegradable and non-toxic.	The sensitivity of polymeric systems in response to changes in temperature varies according to different factors, such as molecular weight and solubility.	[81,86,87]
Phase-sensitive	The drug release rate can be easily modulated through functional modifications or through the use of different solvents or solvent mixtures.	To have a greater effectiveness of the release it is often necessary to associate to these systems some light sensitive components. Most of these systems are not approved by FDA.	[89,90,91]
Light-sensitive	Targeted drug delivery is independent of the conditions of the biological environment. The therapeutic target can be modulated by modulating the wavelength of the radiation used.	Most of these systems are not approved by FDA due to their possible toxicity.	[79,92,93]
Biomolecule-sensitive	These systems have a high specificity.	Biomolecules are substances that are difficult to immobilize in drug delivery systems, which leads to poor control of the release of host species	[80,98]

**Table 2 materials-14-02733-t002:** Different compositions fabricated on different metallic substrates by sol–gel method in corrosive NaCl media.

Sol–Gel Coating	Additive Agent	Substrate	Average Thickness	Ref.
GPTMS/TEOS	none	Carbon Steel	47.6–92.8 μm	[159]
GPTMS/TEOS/TETA	TETA	Mild steel	8–10 μm	[182]
TEOS/APTES	Caffeine	Mild steel	5 μm	[183]
Zeolite	X-type zeolite/polyaniline	Carbon Steel	Not reported	[184]
TEOS/TEMS	MMT + Basil extract	Mild steel	40 μm	[185]
GPTMS/TEOS	Silica NPs	Mild steel	20–40 μm	[186]
GPTMS/TEOS	AP / BPA + TPOZ (Zr^4+^)	Hot dip galvanised steel (HDG)	0.8–2.4 μm	[187]
GPTMS/TEOS	SiO_2_ Ce(NO_3_)_3_	Aluminum Alloy AA7075	4.2–8.6 μm	[188]
DMDES/MTES	Not reported	Stainless steel AISI 304	580–760 μm	[189]
GPTMS/MTEOS	TiO_2_ NPs	Aluminum alloy AA6061–T6	2 μm	[176]
Zeolite / DMDMS/PTMS	None	Aluminum alloy AA6061	15.0 μm	[190]
GPTMS/MTEOS	Graphene Oxide	Aluminum alloy AA6061–T6	200–300 nm	[191]
GPTMS/TEOS + Nanotubes Halloysite	Ce^3+^/Zr^4+^	Magnesium alloys AZ91D	3–3.5 μm	[167]
TEOS/VTMS	EDTPO	Aluminum alloy AA2024–T3	145–200 nm	[192]
GLYMO/AMEO	Not reported	Mild steel	1.8–2 μm	[193]
GPTMS/TEOS	Silica/alumina NPs	Carbon steel	Not reported	[180]
TEOS/ MTES	ZrO_2_ TiO_2_	Stainless steel AISI 304	115–545 μm	[188]

^1^ GPTMS = (3-glycidyloxypropyl)trimethoxysilane; TEOS = tetraethylorthosilane; TETA = triethylenetetramine; APTES = (3-aminopropyl)triethoxysilane; TEMS = teriethoxymethylsilane; TMS = *N*-propyl-trimethoxy-silane; MTES = methyltriethoxysilane; GLYMO = glycidoxypropyltrimethoxysilane; DMDES = dimethyldiethoxysilane; DMDMS = dimethyl-dimethoxy-silane; 1H,1H,2H,2H-Perfluorooctyltriethoxysilane; VTMS = vinyltrimethoxysilane; AMEO = aminopropylethoxysilane; MMT = montmorillonite; AP = mono-phenol; BPA = bi-phenol; TPOZ = zirconium (IV) n-propoxide; EDTPO = ethylenediamine tetra(methylene phosphonic acid).

## Data Availability

Data sharing not applicable.

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
