# Peer review of "Nanostructured Surface Finishing and Coatings: Functional Properties and Applications"

_materials, 2021, doi:10.3390/ma14112733_

Round 1
Reviewer 1 Report
In this review, Ielo, et al. summarized broadly over techniques and experimental results relating to surface coatings. The review is written very thoroughly with a great background introduction and review over the status of understanding. This topic will be interesting to other researchers but there are some comments about the organization of the review. Detailed comments are listed below:
Major comment:
- The connections between different sections are weak. For example, “smart materials” is a very broad topic, and it also overlaps with part of the discussion in drug delivery, such as pH-sensitive materials. Mixing these fields makes it hard for the audience to find the targeted sections.
- It is suggested to break the paragraph from lines 113-212 into several paragraphs. It is very hard to read through and a summarizing figure would be helpful if possible. Similar comments also apply to paragraphs from lines 384-462, lines 1093-1177. On the other hand, paragraphs from lines 1191-1196 are fragmented statements.
- Section 2 on smart materials doesn’t emphasize the “nanostructure” aspect of the coating. The figures included in section 2 looks quite random, and it highlights three work at the end of the paragraph about pH detection, which is not the major points of the section. It is suggested to include a paragraph discussing the relationship with other existing reviews on related topics. Here are a few reviews on smart nanomaterials to my knowledge: Reversibly Reconfigurable Colloidal Plasmonic Nanomaterials, Am. Chem. Soc., 139, 15, 5266–5276 (2017); On-Demand Reconfiguration of Nanomaterials: When Electronics Meets Ionics, Adv. Mater., 30, 1, 1702770 (2017); Reconfigurable Nanoscale Soft Materials, Curr. Opin. Solid State Mater. Sci., 23,1, 41–49 (2019).
Minor comments:
- Line 81, “they” should be “them”.
- Line 203, “NY” seems first used here while the explanation is in line 226. The abbreviation should be explained when first used.
- Line 1178, “5” should be “8”.
Author Response
Major comment:
Comment 1: The connections between different sections are weak. For example, “smart materials” is a very broad topic, and it also overlaps with part of the discussion in drug delivery, such as pH-sensitive materials. Mixing these fields makes it hard for the audience to find the targeted sections.
According to the Reviewer’s comment, and to clarify the connection between paragraphs and their topics, the title of the paragraph “smart materials” was changed and it is now entitled “Advance in halochromic smart textiles”; it has also been divided into subparagraphs, as following:
2.1 smart materials: an introduction;
2.2 Textile-based smart materials;
2.3 Textile-based optical pH sensors;
2.3.1 Strategies for developing halochromic coating for optical textile based pH sensors
2.3.1.1 Electrospinning
2.3.1.2 Grafting polymerization
2.3.1.3 sol-gel technique
Comment 2: It is suggested to break the paragraph from lines 113-212 into several paragraphs. It is very hard to read through and a summarizing figure would be helpful if possible. Similar comments also apply to paragraphs from lines 384-462, lines 1093-1177. On the other hand, paragraphs from lines 1191-1196 are fragmented statements.
Lines 113-212: According to the Reviewer’s comment, the whole paragraph “Advance in halochromic smart textiles” was divided into different subparagraphs (see previous answer).
Lines 384-462: According to the Reviewer’s comment, the paragraph “3. Drug delivery/release nanosystems” has been organized in several subparagraphs, as following:
3.1 Functional coating for drug-delivery nanosystems: an introduction
3.1.1 Self-assembled polymers for nanocarriers
3.1.2 Functional or coated nanofillers
3.2 Smart polymers for drug nanocarriers
3.2.1 Temperature-sensitive smart polymers
3.2.2 Phase-sensitive smart polymers
3.2.4 Biomolecule-sensitive smart polymers
According to the Reviewer’s comment 1, the whole sub-paragraph “3.2.1 pH-sensitive smart polymers” has been deleted.
Lines 1093-1177: According to the Reviewer’s comment, the paragraph “7. Antifouling coatings” has been organized in several subparagraphs, as following:
7.1 Antifouling solutions tributyltin -based
7.2 Eco-friendly silica-based AF formulations
7.3 Nanostructured biocide-based AF formulations
7.4 Hydrorepellent coatings
Lines 1191-1196: According to the Reviewer’s comment, the text has been rearranged as following “As a matter of fact, in wastewater treatment the simplest and most efficient methodology to modify the surface properties of filtering membranes, thus improving their retention and regeneration properties, involves the use of membrane coatings based on nanomaterials.
The recently described literature also highlights the characteristic proprieties of sol–gel polymeric materials and their possible employment in the development of wide-employed 3D matrices for the development of useful coatings for corrosion mitigation on different metallic surfaces or structural consolidation, as well as smart, fire-resistant, and antifouling silica-based coatings.”
Comment 3: Section 2 on smart materials doesn’t emphasize the “nanostructure” aspect of the coating. The figures included in section 2 looks quite random, and it highlights three work at the end of the paragraph about pH detection, which is not the major points of the section. It is suggested to include a paragraph discussing the relationship with other existing reviews on related topics. Here are a few reviews on smart nanomaterials to my knowledge: Reversibly Reconfigurable Colloidal Plasmonic Nanomaterials, Am. Chem. Soc., 139, 15, 5266–5276 (2017); On-Demand Reconfiguration of Nanomaterials: When Electronics Meets Ionics, Adv. Mater., 30, 1, 1702770 (2017); Reconfigurable Nanoscale Soft Materials, Curr. Opin. Solid State Mater. Sci., 23,1, 41–49 (2019).
According to the Reviewer’s comments:
- a sentence to emphasize the “nanostructure” aspect of the coating was added in paragraph 2;
- figures in the whole section were not modified since the title of paragraph 2 was changes to focus the topic onto halochromic smart textiles as a pH sensitive device;
- A paragraph dealing with plasmonic nanoparticles following the references as suggested by the Reviewer was added.
Minor comments:
- Line 81, “they” should be “them”.
According to the Reviewer’s comment, “they” in line 31 was changed in “them”
- Line 203, “NY” seems first used here while the explanation is in line 226. The abbreviation should be explained when first used.
According to the Reviewer’s comment, the abbreviation “NY” was introduced when first used in the text.
- Line 1178, “5” should be “8”.
According to the Reviewer’s comment, “5” has been replaced with “8”
Reviewer 2 Report
The manuscript of review article entitled “Nanostructured surface finishing and coatings: functional properties and applications” should be reject by the editorial office of Materials. The authors aim to elucidate an overview of nanostructured coatings according to functional properties and related applications. There are references were list up to 231 articles but poor organized. Especially, smart materials and drug release/delivery systems relies on their intrinsic properties in section 2 and 3, not their surface properties nor their nanostructure. Section 4 to 7 would be the key in accordance with the manuscript title. After all, several suggestions should be noted to revise the manuscript if authors want to know more about smart materials and drug delivery systems.
Suggestions are addressed below:
(1) In the line 43, grain boundaries would be a defect in the structure so as to deteriorate materials property expect particular cases. Also, the introduction section doesn’t help reviewer to know better manuscript’s topic. Similar situation can be observed in the whole article.
(2) The author describes reference no.15 in the page 3 and 4. The main idea is obscure and hard to understand the logic thinking.
(3) In Section 3, authors quote research results from Poly (N-isopropyl acrylamide) (PNIPAAm) to introduce temperature-sensitive drug delivery system by the critical micellar concentration. Another famous system would be Pluronic block copolymers, its research should be included in this section.
(4) In the line 226, the abbreviation of NY dyestuff and PA should be defined at the first time in the article. By the way, the concept of textile polymers and textilification should be different to smart materials.
(5) For the reference no.112 and no.113, the current developing trend would focus on non-halogen based compounds, however, the paragraph discusses less about this idea and possible mechanism. But, authors clearly mentioned phosphorus based, nitrogen based, silicon based compounds and minerals for the applications in flame retardant coatings. It is good point.
Author Response
Review #2:
The manuscript of review article entitled “Nanostructured surface finishing and coatings: functional properties and applications” should be reject by the editorial office of Materials. The authors aim to elucidate an overview of nanostructured coatings according to functional properties and related applications. There are references were list up to 231 articles but poor organized. Especially, smart materials and drug release/delivery systems relies on their intrinsic properties in section 2 and 3, not their surface properties nor their nanostructure. Section 4 to 7 would be the key in accordance with the manuscript title. After all, several suggestions should be noted to revise the manuscript if authors want to know more about smart materials and drug delivery systems.
According to the Reviewer’s comments, a sentence to emphasize the “nanostructure” aspect of the coating was added in paragraph 2. Moreover, to make the paragraph 2 more readable, the same is now entitled “Advance in halochromic smart textiles” and divided into different subparagraphs focusing on pH-sensitive smart textiles.
Suggestions are addressed below:
- In the line 43, grain boundaries would be a defect in the structure so as to deteriorate materials property expect particular cases. Also, the introduction section doesn’t help reviewer to know better manuscript’s topic. Similar situation can be observed in the whole article.
According to the Reviewer’s comment, in the line 43 the sentence “Most of the atoms are associated with grain boundaries for contributing to improving the physical properties of nanostructured materials. “ was changed with the following sentence: “So most of the atoms, by aggregating, contribute to change the size of the grain boundaries thus improving the physical properties of nanostructured materials; this in according to the Hall - Petch reinforcement method, by which the change in grain size can affect the number of dislocations accumulated at the grain boundary and the yield strength. [2].”
- The author describes reference no.15 in the page 3 and 4. The main idea is obscure and hard to understand the logic thinking.
According to the Reviewer’s comment, the text in page 3 and 4 discussing reference [15] was deleted.
(3) In Section 3, authors quote research results from Poly (N-isopropyl acrylamide) (PNIPAAm) to introduce temperature-sensitive drug delivery system by the critical micellar concentration. Another famous system would be Pluronic block copolymers, its research should be included in this section.
According to the Reviewer’s comment, the following sentence was added: “Pluronic block copolymer is a thermosensitive polymer capable of releasing biomolecules (proteins or lipoproteins) as a function of temperature variations. Pluronic diblock copolymer microspheres have a porous and hydrophilic structure. The release rate of the biomolecules is modulated according to the temperature variations and the surface functionalization of the Pluronic copolymer.”
- In the line 226, the abbreviation of NY dyestuff and PA should be defined at the first time in the article. By the way, the concept of textile polymers and textilification should be different to smart materials.
According to the Reviewer’s comment, the abbreviation “NY” and “PA” were introduced when first used in the text. “Textile polymers” was changed in “textile fabrics”. The sentence containing the definition of “textilification” was slightly changed to avoid misunderstanding and make the generic term more in agreement with the context.
(5) For the reference no.112 and no.113, the current developing trend would focus on non-halogen based compounds, however, the paragraph discusses less about this idea and possible mechanism. But, authors clearly mentioned phosphorus based, nitrogen based, silicon based compounds and minerals for the applications in flame retardant coatings. It is good point.
In the paper, the current developing trend for environmentally friendly flame retardant treatments was specifically presented. Besides phosphorus-, nitrogen- and silicon-based compounds, in the field of nanotechnology applied to improve flame retardancy of materials, two approaches were shown (references [134][135]): the addition of nanosized particles in traditional back-coatings and the deposition of nanosized films.
Finally, to better explain and focus on the subject matter, two new figures (with captions) have been inserted, i.e. figure 1 in paragraph 2.1 and figure 6 in paragraph 4.
Reviewer 3 Report
The manuscript by Ielo et al. offered a detailed overview of the properties and applications of the nanostructured surfaces and coatings, with the focus on the smart material, drug delivery systems, industrial, antifouling, and nano/ultrafiltration membrane coatings. The authors provide current progress on the synthesis and consequent physical/chemical properties of each coating. The article highlights the important characteristics of each coating and also the key parameters. This paper gives an interesting scientific perspective on a field that has recently advanced. I found this paper an approachable yet in-depth review for the general audience.
I would recommend this work for publication, but I also suggest the authors consider addressing some minor concerns for improvement, listed below.
1) In section 3.4, the authors only briefly touch on the light-sensitive smart polymers by commenting on the general definition of the light-sensitive smart polymers. Since this type of material has its own unique advantages, can authors comment on other advantages and disadvantages? Such as the dark toxicity?
2) Can authors comment on the comparison between the UV, visible, and near IR-sensitive polymers? Below are several journal articles I found on the topic of light-sensitive polymers, and I think these papers would be a great addition to the review paper.
i) You, Jian, et al. "Near‐infrared light triggers release of paclitaxel from biodegradable microspheres: photothermal effect and enhanced antitumor activity." Small9 (2010): 1022-1031.
3) I recommended adding a table under section 3 to summarize the various smart polymer platforms, to clearly and concisely show the reader the advantages and limitations of each type (Temperature, pH, light, phase, biomolecule)
4) figure 6 is not clear, four arrows are pointing upward, but no labels. The meaning of the labels on the bottom of the figure “biodegradation” and “total mineralization” was not clearly stated. Could authors improve this figure?
5) To give the audience a complete picture of the smart material, the phase-change material in response to the temperature could also be categorized as volatile and non-volatile phase change materials. The volatile phase change materials, their resistivity, and optical properties change based on the temperature and this change is reversible. For the non-volatile phase change material, based on the temperature, its phase is not reversible automatically but needs an external excitation. Can authors add these types of material in the smart material section? Below is the material on recent key advances of the corresponding material fabrication and properties, which I think would be a great addition to the review.
i) https://doi.org/10.1016/j.ceramint.2016.10.091
ii) https://doi.org/10.3390/app9030530
Author Response
Review #3:
The manuscript by Ielo et al. offered a detailed overview of the properties and applications of the nanostructured surfaces and coatings, with the focus on the smart material, drug delivery systems, industrial, antifouling, and nano/ultrafiltration membrane coatings. The authors provide current progress on the synthesis and consequent physical/chemical properties of each coating. The article highlights the important characteristics of each coating and also the key parameters. This paper gives an interesting scientific perspective on a field that has recently advanced. I found this paper an approachable yet in-depth review for the general audience.
I would recommend this work for publication, but I also suggest the authors consider addressing some minor concerns for improvement, listed below.
1) In section 3.4, the authors only briefly touch on the light-sensitive smart polymers by commenting on the general definition of the light-sensitive smart polymers. Since this type of material has its own unique advantages, can authors comment on other advantages and disadvantages? Such as the dark toxicity?
2) Can authors comment on the comparison between the UV, visible, and near IR-sensitive polymers? Below are several journal articles I found on the topic of light-sensitive polymers, and I think these papers would be a great addition to the review paper.
- i) You, Jian, et al. "Near‐infrared light triggers release of paclitaxel from biodegradable microspheres: photothermal effect and enhanced antitumor activity." Small9 (2010): 1022-1031.
According to the Reviewer’s comments 1 and 2 the light-sensitive smart polymers paragraph has been implemented with the following text: “These light-sensitive systems have the advantage of delivering drugs to a specific site independently of the conditions of the biological environment [92]. The study of the photo-regulation mechanism of these systems and the development of new biocompatible materials for in vivo applications in drug administration [93]. Electromagnetic radiation, in the wavelength range between 2500-380 nm, applied in vivo can activate and deactivate the release of the biomolecule or drug in a specific organ or tissue, allowing excellent control of the release and reducing damage to adjacent sites [94]. UV light acts as a trigger for topical treatments [95], as the radiation used (λ < 700 nm) undergoes absorption by endogenous biomolecules and is therefore unable to penetrate more than 10 mm deep into the tissue [96]. UV treatment is therefore limited to therapies on the superficial layers of the skin or some internal organs. To obtain a slightly deeper tissue penetration, near infrared (NIR) light was used in the wavelength range from 650 to 900 nm, in this area of the spectrum the endogenous molecules have a minimal absorption of light, reducing the interference with the tissue. NIR imaging techniques are non-invasive in vivo methods that allow to visualize physiological and metabolic processes [97]. Light-induced therapies fall into two categories, photodynamic therapy in which light stimulates apoptosis or necrosis by administering a photosensitizer that reacts with the light and oxygen present in the tissue to produce singlet oxygen. The second approach is photo-polymerization which induces the in situ formation of filling materials. (33) used for the preparation of dental composites or implants that take the shape of the implant and are applied without the use of invasive methods. Some light-sensitive systems that use azobenzene groups and similar aromatic substances are listed among the toxic compounds by the FDA, which limits their clinical use and pushes researchers to study alternative biocompatible materials [79].”
3) I recommended adding a table under section 3 to summarize the various smart polymer platforms, to clearly and concisely show the reader the advantages and limitations of each type (Temperature, pH, light, phase, biomolecule)
According to the Reviewer’s comment Table 1 has been added which summarizes the advantages and disadvantages of using the various smart polymer platforms
4) figure 6 is not clear, four arrows are pointing upward, but no labels. The meaning of the labels on the bottom of the figure “biodegradation” and “total mineralization” was not clearly stated. Could authors improve this figure?
According to the Reviewer’s comment the figure has been changed (now renumbered as Fig. 8).
5) To give the audience a complete picture of the smart material, the phase-change material in response to the temperature could also be categorized as volatile and non-volatile phase change materials. The volatile phase change materials, their resistivity, and optical properties change based on the temperature and this change is reversible. For the non-volatile phase change material, based on the temperature, its phase is not reversible automatically but needs an external excitation. Can authors add these types of material in the smart material section? Below is the material on recent key advances of the corresponding material fabrication and properties, which I think would be a great addition to the review.
- i) https://doi.org/10.1016/j.ceramint.2016.10.091
- ii) https://doi.org/10.3390/app9030530
As suggested by the Reviewer, a paragraph dealing with non-volatile reversible phase change materials following the references was added.
Round 2
Reviewer 2 Report
According to Authors' reply, I would recommend this manuscript for publication. After i carefully read the other two reviewer's previous comments, the authors rewrite original manuscript and make it better. Also, my concerns were fully answered. This review article makes progress in the material research.